# OBJECT-CENTRIC ARCHITECTURES ENABLE EFFICIENT CAUSAL REPRESENTATION LEARNING

**Amin Mansouri***
Mila, Quebec AI Institute

**Jason Hartford***
Valence Labs

**Yan Zhang**
Samsung - SAIT AI Lab, Montreal

**Yoshua Bengio**
Mila, Quebec AI Institute / U. Montreal / CIFAR

## ABSTRACT

Causal representation learning has showed a variety of settings in which we can disentangle latent variables with identifiability guarantees (up to some reasonable equivalence class). Common to all of these approaches is the assumption that (1) the latent variables are represented as $d$-dimensional vectors, and (2) that the observations are the output of some injective generative function of these latent variables. While these assumptions appear benign, we show that when the observations are of multiple objects, the generative function is no longer injective and disentanglement fails in practice. We can address this failure by combining recent developments in object-centric learning and causal representation learning. By modifying the Slot Attention architecture (Locatello et al., 2020b), we develop an object-centric architecture that leverages weak supervision from sparse perturbations to disentangle each object's properties. This approach is more data-efficient in the sense that it requires significantly fewer perturbations than a comparable approach that encodes to a Euclidean space and we show that this approach successfully disentangles the properties of a set of objects in a series of simple image-based disentanglement experiments.

## 1 INTRODUCTION

Consider the image in Figure 1 (left). We can clearly see four different colored balls, each at a different position. But asking, "Which is the first shape? And which is the second?" does not have a clear answer: the image just depicts an unordered set of objects. This observation seems trivial, but it implies that there exist permutations of the objects which leave the image unchanged. For example, we could swap the positions of the two blue balls without changing a single pixel in the image.

In causal representation learning, the standard assumption is that our observations $x$ are "rendered" by some generative function $g(\cdot)$ that maps the latent properties of the image $z$ to pixel space (i.e. $x = g(z)$); the goal is to *disentangle* the image by finding an "inverse" map that recovers $z$ from $x$ up to some irrelevant transformation. The only constraint on $g(\cdot)$ that is assumed by all recent papers (for example Hyvarinen & Morioka, 2016; 2017; Locatello et al., 2020a; Khemakhem et al., 2020a;b; Lachapelle et al., 2022; Ahuja et al., 2022a;b; 2023), is that $g(\cdot)$ is injective[1], such that $g(z_1) = g(z_2)$ implies that $z_1 = z_2$. But notice that if we represent the latents $z$ as some $d$-dimensional vectors in Euclidean space, then whenever we observe objects like those shown in Figure 1, this injectivity assumption fails: symmetries in the objects' pixel representation imply that there exist non-trivial permutation matrices $\Pi$, such that $g(z) = g(\Pi z)$. This is not just a theoretical inconvenience: Figure 1 (right) shows that when the identity of the balls is not distinguishable, the disentanglement performance of a recent approach from Ahuja et al. (2022b) is upper-bounded by $1/k$ where $k$ is the number of balls.

---

*correspondence to `amin.mansouri@mila.quebec` or `jason@valencelabs.com`

[1]Some papers place stronger constraints on $g(\cdot)$, such as linearity Hyvärinen & Oja, 2000; Squires et al., 2023, sparsity Moran et al., 2022; Zheng et al., 2022, or constraints on $g$'s Jacobian Gresele et al., 2021; Brady et al., 2023 but injectivity is the weakest assumption common to all approaches.

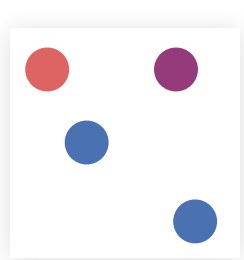 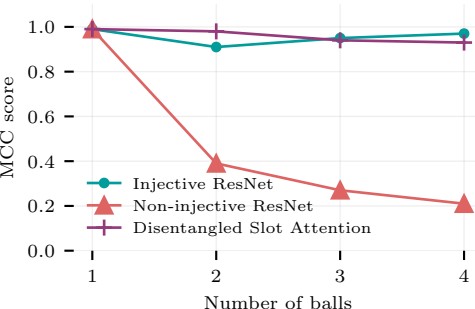

Figure 1: *(Left)* An example image of simple objects. *(Right)* Mean correlation coefficient (MCC) score which measures the correlation between inferred latent variables and their associated ground truth values. Ahuja et al. (2022b)'s approach achieves almost perfect MCC scores (i.e. a score $\approx 1$) when the ball color is used to make the generative function injective ("Injective ResNet"), but achieves an MCC score of at most $\frac{1}{k}$ where $k$ is the number of objects when colors are selected randomly ("Non-injective ResNet"). We show that it is possible to recover the injective performance by disentangling object-centric representations ("Disentangled Slot Attention").

In parallel to this line of work, there has been significant progress in the object-centric learning literature (e.g. van Steenkiste et al., 2018a; Goyal et al., 2019; Locatello et al., 2020b; Goyal et al., 2020; Lin et al., 2020; Zhang et al., 2023) that has developed a suite of architectures that allow us to separate observations into sets of object representations. Two recent papers (Brady et al., 2023; Lachapelle et al., 2023) showed that the additive decoders used in these architectures give rise to provable object-wise disentanglement, but they did not address the task of disentangling the objects' associated properties. In this paper we show that by leveraging object-centric architectures, *we effectively reduce the multi-object problem to a set of single-object disentanglement problems* which not only addresses injectivity failures, but also results in a significant reduction in the number of perturbations we need to observe to disentangle properties using Ahuja et al. (2022b)'s approach. We illustrate these results by developing a property disentanglement algorithm that combines Zhang et al. (2023)'s SA-MESH object-centric architecture with Ahuja et al. (2022b)'s approach to disentanglement and show that our approach is very effective at disentangling the properties of objects on both 2D and 3D synthetic benchmarks.

In summary, we make the following contributions:

- We highlight two problems that arise from objects which violate standard assumptions used to identify latent variables (Section 3).
- We show that these problems can be addressed by leveraging object-centric architectures, and that using object-centric architectures also enables us to use a factor of $k$ fewer perturbations to disentangle properties, where $k$ is the number of objects (Section 4).
- We implement the first object-centric disentanglement approach that disentangles object properties with identifiability guarantees (Section 5).
- We achieve strong empirical results[2] on both 2D and 3D synthetic benchmarks (Section 7).

## 2 BACKGROUND

Causal representation learning (Schölkopf et al., 2021) seeks to reliably extract meaningful latent variables from unstructured observations such as images. This problem is impossible without additional structure because there are infinitely many latent distributions $p(z)$ that are consistent with the observed distribution, $p(x) = \int p(x|z)dp(z)$, only one of which corresponds to the ground truth distribution (Hyvärinen & Pajunen, 1999; Locatello et al., 2019). We therefore need to restrict the solution space either through distributional assumptions on the form of the latent distribution $p(z)$, or through assumptions on the functional form of the generative function $g : \mathcal{Z} \to \mathcal{X}$ that maps from the latent space to the observed space (Xi & Bloem-Reddy, 2023). A key assumption that (to the

---

[2]The code to reproduce our results can be found at: https://github.com/amansouri3476/OC-CRL

best of our knowledge) is leveraged by all papers that provide identifiability guarantees, is that $g(\cdot)$ is injective such that if we see identical images, the latents are identical (i.e. if $g(z_1) = g(z_2)$ then $z_1 = z_2$).

Given these restrictions, we can analyze the *identifiability* of latent variables for a given inference algorithm by considering the set of optimal solutions that satisfy these assumptions. We say latent variables are *identified* if the procedure will recover the latents exactly in the infinite data limit. Typically, some irreducible indeterminacy will remain, so latent variables will be identified *up to* some equivalence class $\mathcal{A}$. For example, if the true latent vector is $z$, and we have an algorithm for which all optimal solutions return a linear transformation of $z$ such that, $\mathcal{A} = \{A : \hat{z} = Az\}$, then we say the algorithm is linearly identifies latent variables. We will call latent variables *disentangled* if the learning algorithm recovers the true latent variables up to a permutation (corresponding to a relabeling of the original variables), and element-wise transformation. That is, for all $i$, $z_i = h_i(z_{\pi(i)})$, where $\pi$ is a permutation, and $h_i(\cdot)$ is an element-wise function; for the results we consider in this paper this function is simply a scaling and offset, $f_i(z) = a_i z_i + b_i$ corresponding to a change of units of measurement and intercept.

In this paper, we will build on a recent line of work that leverages paired samples from sparse perturbations to identify latent variables (Locatello et al., 2020a; Brehmer et al., 2022; Ahuja et al., 2022b). Our approach generalizes Ahuja et al. (2022b) to address the non-injectivity induced by objects, so we will briefly review their main results. Ahuja et al. assume that they have access to paired samples, $(x, x')$ where $x = g(z)$, $x' = g(z')$, and $z_i$ is perturbed by a set of sparse offsets $\Delta = \{\delta_1, \ldots, \delta_k\}$, such that $z_i' = z_i + \delta_i$ for all $i \in \{1, \ldots, k\}$. They show that if $g(\cdot)$ is an injective analytic function from $\mathbb{R}^d \to \mathcal{X}$, every $\delta \in \Delta$ is 1-sparse, and at least $d$ linearly independent offsets are observed, then an encoder, $f$ that minimizes the following objective recovers the true $z$ up to permutations, scaling and an offset (Ahuja et al., 2022b, Theorem 1),

$$\hat{f} \in \arg\min_{f'} E_{x,x',\delta} \left[ (f'(x) + \delta - f'(x'))^2 \right] \quad \Rightarrow \quad \hat{f}(x) = \hat{z} = \Pi\Lambda z + c \tag{1}$$

where $\Pi$ is a permutation matrix, $\Lambda$ is an invertible diagonal matrix and $c$ is an offset.

## 3 OBJECTS RESULT IN NON-IDENTIFIABILITY

We begin by formally characterizing the challenges that arise when images contain multiple objects.

**Data generating process.** We assume that a set $Z := \{z_i\}_{i=1}^k$ of $k$ objects is drawn from some joint distribution, $\mathbb{P}_Z$. In order to compare set and vector representations, let $\textbf{vec}_\pi(Z)$ denote a flattened vector representation of $Z$ ordered according to some permutation $\pi \in \text{Sym}(k)$, the symmetric group of permutations of $k$ objects; when $\pi$ is omitted, $\textbf{vec}(Z)$ simply refers to an arbitrary default ordering (i.e. the identity element of the group). Each object is described by a $d$-dimensional vector of properties[3] $z_i \in \mathbb{R}^d$, and hence $\textbf{vec}(Z) \in \mathbb{R}^{kd}$. We say objects have *shared properties* if the coordinates of $z_i$ have consistent meaning across objects. For example, the objects in Figure 1 (left), each have $x, y$ coordinates and a color which can be represented by its hue, so $z_i = [p_x^i, p_y^i, h^i]$. In general, the set of properties associated to an object can be different across objects, but for simplicity, our discussion will focus on properties that are fully shared between all objects.

**The non-injectivity problem.** We observe images $x$ which are generated via a generative function $\mathfrak{g}(\cdot)$ that renders a set of object properties into a scene in pixel space, such that $x = \mathfrak{g}(Z)$. While $\mathfrak{g}(\cdot)$ is a set function, we can define an equivalent vector generative function, $g$, which, by definition, produces the same output as $\mathfrak{g}(Z)$; i.e. for all $\pi \in \text{Sym}(k)$, $g(\textbf{vec}_\pi(Z)) = \mathfrak{g}(Z)$. This generative function $g$ taking vectors as input is consistent with standard disentanglement assumptions except that it is not injective:

**Proposition 1.** *If $g(\textbf{vec}_\pi(Z)) = \mathfrak{g}(Z)$ for all $\pi \in \text{Sym}(k)$, then $g(\cdot)$ is not injective.*

---

[3] A natural extension of the perspective we take in this paper is to also treat properties as sets rather than ordered vectors; for example, see Singh et al. (2023). We leave understanding the identifiability of these approaches to future work.

*Proof.* The contrapositive of the definition of injectivity states that $z_1 \neq z_2$ implies $g(z_1) \neq g(z_2)$, but by definition of $g(\cdot)$, there exist $z_1 \neq z_2$ such that $g(z_1) = g(z_2)$. In particular, for any set $Z$ and permutations $\pi_1 \neq \pi_2 \in \mathrm{Sym}(k)$, the vectors $\mathbf{vec}_{\pi_1}(Z) = z_1 \neq z_2 = \mathbf{vec}_{\pi_2}(Z)$. $\qquad\square$

This proposition simply states that if images are composed of *sets* of objects, then if we model the generative function as a map from a Euclidean space, this map will not be injective by construction.

With the exception of Lachapelle et al. (2023), all of the causal representation learning papers cited in section 6 assume the generative function $g$ is injective. To see why injectivity is necessary in general, consider an image with two objects. If the two objects are identical, then there are two disentangled solutions corresponding to the two permutations, so it is not possible to identify a unique solution.

**The object identity problem.** When applying sparse perturbations on $Z$ (see section 2), we are effectively perturbing one coordinate of one object. However, how can we know which object of the multiple possible objects in $Z$ we have perturbed? In the case of injective mappings, this is simple: since there is a consistent ordering for them, we know that a coordinate in $\mathbf{vec}(Z)$ corresponds to the same object before and after the perturbation.

However, this is no longer the case in our setting. Since the objects are actually part of a set, we cannot rely on their ordering: the perturbed object can, in principle, freely swap order with other objects; there is no guarantee that the ordering before and after the perturbation remains the same. In fact, we know that these ordering changes *must* be present due to the *responsibility problem*:

**Proposition 2** (Zhang et al. (2020); Hayes et al. (2023)). *If the data is generated according to the data generating process described above with $g(\mathbf{vec}_\pi(Z)) := \mathfrak{g}(Z)$ and $k > 1$, then $f(\cdot)$ is discontinuous.*

*Proof Sketch.* Consider Figure 2, notice that if we perform a $90°$ rotation in pixel space of the image, the image is identical, but the latent space has been permuted, since each ball has swapped positions. Because the image on the left and the image on the right are identical in pixel space, any encoder, $f : \mathcal{X} \to \mathbb{R}^{kd}$, will map them them to identical latents. There exists a continuous pixel-space rotation from $0°$ to $90°$, but it must entail a discontinous swap in which latent is *responsible* for which part of pixel-space according to the encoder. $\qquad\square$

A general proof can be found in Hayes et al. (2023). These discontinuities manifest themselves as changes in permutation from one $\mathbf{vec}_{\pi_1}(Z)$ to another $\mathbf{vec}_{\pi_2 \neq \pi_1}(Z)$. In disentanglement approaches that leverage paired samples (e.g. Ahuja et al., 2022b; Brehmer et al., 2022), continuity enables the learning algorithm to implicitly rely on the object identities to stay consistent. Without continuity, one cannot rely on the fact that $\mathbf{vec}(Z)$ and $\mathbf{vec}(Z) + \delta$ should be the same up to the perturbation vector $\delta$, because the perturbation may result in a discontinuous change of $\mathbf{vec}(Z) + \delta$ when an observation is encoded back to latent space. As a consequence, we lose track of which object we have perturbed in the first place, so naïve use of existing disentanglement methods fails.

Another challenge is that the encoder $f$ (Equation 1) has to map observations to $\mathbf{vec}(Z)$ in a *discontinuous* way, which is traditionally difficult to model with standard machine learning techniques.

In summary, the unordered nature of objects in $Z$ results in non-injectivity, losing track of object identities, and the need for learning discontinuous functions. These all contribute to the non-identifiability of traditional disentanglement methods in theory and practice.

## 4 OBJECT-CENTRIC CAUSAL REPRESENTATION LEARNING

A natural solution to this problem is to recognize that the latent representations of multi-object images are sets and should be treated as such by our encoders and decoders in order to enforce invariance among these permutations. Both Brady et al. (2023) and Lachapelle et al. (2023) showed that architectures that enforce an appropriate object-wise decomposition in their decoders *provably* disentangle images into object-wise blocks of latent variables. These results do not disentangle the properties of objects, but they solve an important precursor: the assumption that there exists an object-wise decomposition of the generative function is sufficient to partition the latents into objects.

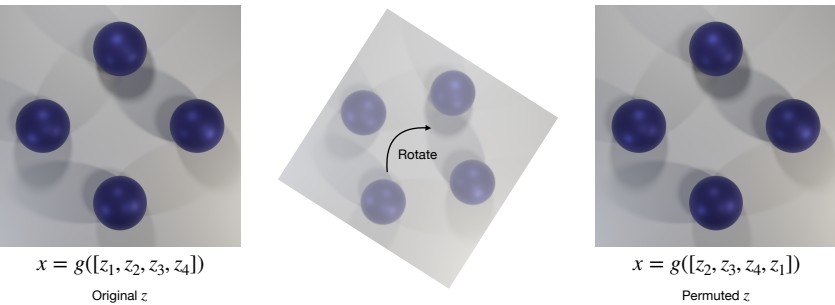

$x = g([z_1, z_2, z_3, z_4])$

Original $z$

$x = g([z_2, z_3, z_4, z_1])$

Permuted $z$

Figure 2: An illustration of the object identity problem. Permuting the order of the latents $[z_1, z_2, z_3, z_3]$ is equivalent to a 90 degree rotation in pixel-space.

Like these two papers, we will assume that natural images can be decomposed into objects,[4] each of which occupies a disjoint set of pixels. When this is the case, we say that an image is *object-separable*. To define object separability formally, we will need to consider a partition $P$ of an image into $k$ disjoint subsets of pixels $P = \{x^{(1)}, \ldots, x^{(k)}\}$ indexed by an index set $\mathcal{I}_P = \{1, \ldots, k\}$; further, denote an index set that indexes the set of latent variables $Z$ as $\mathcal{I}_Z$. We can then say,

**Definition 1.** *An image, $x$, is **object-separable** if there exists an **object-wise** partition $P$ and a bijection $\sigma : \mathcal{I}_P \to \mathcal{I}_Z$ that associates each subset of pixels in $P$ with a particular element of the set of latents, $z_i$, such that each subset of pixels $x^{(i)} \in P$ is the output of an injective map with respect to its associated latent $z_{\sigma(i)}$. That is, for all $i$, $(x^{(i)'} \subset \mathfrak{g}(Z'), x^{(i)} \subset \mathfrak{g}(Z))$, we have that $x^{(i)'} = x^{(i)}$ implies $z'_{\sigma(i)} = z_{\sigma(i)}$.*

This definition says that an image can be separated into objects if it can be partitioned into parts such that each part is rendered via an injective map from some latent $z_i$. We can think of each $x^{(i)}$ as a patch of pixels, with a bijection $\sigma$ that relates each of the $k$ patches of pixels in the partition $\{x^{(i)}\}_{i=1}^k$ to a latent variable in $Z = \{z_i\}_{i=1}^k$. Each patch "depends" on its associated latent via an injective map.

Brady et al. (2023) and Lachapelle et al. (2023) give two different formal characterizations of partitions $P$ that are consistent with our object-wise definition. Brady et al.'s characterization requires that a differentiable generative function $\mathfrak{g}$ is *compositional*, in the sense that each $x^{(i)} \in P$ only functionally depends[5] on a single $z_j \in Z$, and *irreducible* in the sense no $x^{(i)} \in P$ can be further decomposed into non-trivial subsets that have functionally independent latents. Lachapelle et al.'s assumption is weaker than ours in that they only require that the generative function is defined as $\mathfrak{g}(Z) = \sigma(\sum_{z_i \in Z} g_i(z_i))$ where $\sigma$ is an invertible function, and that $\mathfrak{g}$ is a diffeomorphism that is "sufficiently nonlinear" (see Assumption 2 Lachapelle et al., 2023); object-separable images are a special case with $\sigma$ as the identity function and each $g_i(\cdot)$ rendering a disjoint subset of $x$, and hence their results apply to our setting.

**Disentangling properties with object-centric encoding.** In section 3 we showed that the assumptions underlying sparse perturbation-based disentanglement approach are violated in multi-object scenes. But, the results from Brady et al. (2023) and Lachapelle et al. (2023) show that the objects can be separated into disjoint (but entangled) sets of latent variables. This suggests a natural approach to disentangling properties in multi-object scenes:

- we can reduce the multi-object disentanglement problem to a single-object problem with an object-wise partition of the image. Within each patch of pixels $x^{(i)} \in P$ injectivity holds, and so we no longer have multiple solutions at a patch level. This partition is identifiable and we

---

[4]This is a pragmatic approximation that suffices for the purposes of this paper, but a careful treatment of objects is far more subtle because what we interpret as an "object" often depends on a task or a choice of hierarchy; for a more nuanced treatment, Smith (2019)'s Chapter 8 is an excellent introduction into the subtleties around demarcating an "object".

[5]Functional dependence is defined by non-zero partial derivatives, i.e. $\frac{\partial x^i}{\partial z_j} \neq 0$.

can use an object-centric architecture to learn the object-wise partition. We require that this object-centric architecture can handle the responsibility problem.

- we leverage Ahuja et al. (2022b)'s approach to using weak supervision to disentangle the properties of each object individually. Since we assume that properties between objects are shared, this requires a factor of $k$ fewer perturbations in the perturbation set $\Delta$, where $k$ the number of objects.

- we address the object identity problem where we lose track of object identities after perturbations through an explicit matching procedure that re-identifies the object being perturbed.

See section 5 for details of how we implement this. This approach not only addresses the challenges outlined in Section 3, it also significantly reduces the number of perturbations that we have to apply in order to disentangle shared properties.

**Theorem 1** (informal). *If a data generating process outputs observations with $k$ objects that have shared properties, then an object-centric architecture of the form $F(x) := \{\mathfrak{f}(x^{(i)})\}_{x^{(i)} \in P}$ where $P$ is an object-wise partition and $\mathbf{f} : \mathcal{X} \to \mathbb{R}^d$ will disentangle in $k$ times fewer perturbations than an encoder of the form $f : \mathcal{X} \to \mathbb{R}^{kd}$.*

The proof is given in Appendix A. The main insight is that if we have an object-centric architecture that learns an object-wise partition $P$ and uses the same encoding function $\mathfrak{f}$ on every patch, then every perturbation provides weak supervision to every object, despite the fact that only one was perturbed. As a result, we do not need to disentangle each object's properties separately, and hence we reduce the number of required interventions by a factor of $k$.

## 5 METHOD

**Object-wise partitions**   There exist a number of ways to decompose an image into objects, but for our purposes, pixel segmentation-based approaches (Greff et al., 2019; Locatello et al., 2020b; Zhang et al., 2023) let us directly adapt existing disentanglement techniques to work with object-centric encoders. A pixel segmentation encoder $\hat{f}$ maps from images $x$ to a set of slot vectors $\{s_1, \ldots, s_k\}$, each of which depends on a subset of the pixels $x^{(i)} \in P$. Images are then reconstructed using a slot decoder $\hat{g}$ that maps from the set of slot representation back to pixel space. The dependence between slots and patches of pixels is typically controlled by a soft-attention matrix, which will typically not result in a partition of the pixels. In our implementation, we use Zhang et al.'s SA-MESH modification of the original Locatello et al. slot attention architecture, which adds an entropy regularization term to learn sparse attention matrices that do approximately partition the input by encouraging the subsets of pixels $x^{(i)}$ to be disjoint (for details on the architectures, see Appendix B). Importantly for us, Zhang et al. (2023) is *exclusively multiset-equivariant* (Zhang et al., 2022), which allows it to model discontinuous functions, thus handling the responsibility problem.

Slot attention is usually trained with a reconstruction loss from relatively high-dimensional per-object slot representations, $s_i \in \mathbb{R}^D$, but for the images that we work with, we want a relatively low dimensional latent description (in the simplest case, just the two dimensions representing the $(x, y)$ coordinates of each object). To disentangle these high-dimensional slot representations, we simply add a projection head, $\hat{p} : s_i \to \hat{z}_i$, that is trained by a latent space loss.

**Disentanglement via weak supervision with matching**   Ahuja et al. assume access to pairs of images $(x, x')$ that differ by a sparse offset $\delta$. They enforce this assumption via a disentanglement loss that requires that the latent representations of this pair of images differ by $\delta$, such that $\hat{f}(x) + \delta = \hat{f}(x')$. When using a slot attention architecture, we introduce a *matching step* to the loss to infer the object to which the offset $\delta$ was applied. With 1-sparse $\delta$ vectors, the matching step reduces to a simple minimization over a cost matrix that measures $\|\hat{z}(x'^{(j)}) - (\hat{z}(x^{(i)}) + \delta)\|^2$ for all pairs of slots $i, j$. In Appendix D, we provide a more general matching procedure that applies to settings with dense offsets $\delta$. We jointly optimize the following reconstruction and disentanglement loss,

$$\hat{f}, \hat{g}, \hat{p} \in \underset{f,g,p}{\arg\min} \, E_x[\|x - g(f(x))\|^2] + E_{x,x',\delta}[\min_{i,j} \|(p(f(x')^{(j)}) - (p(f(x)^{(i)}) + \delta_t)\|^2] \quad (2)$$

The first term in this loss enforces that the encoder / decoder pair $\hat{f}, \hat{g}$ capture enough information in the slot representations $s_i$ to reconstruct $x$. The second term contains the matching term and ensures that the function that projects from slot representation to latents $\hat{p}$ disentangles the slot representations into individual properties. The offset $\delta$ could be known or unknown to the model, and for the remainder of this paper, we focus on the more challenging and natural case of unknown offsets. See appendix C for more details.

## 6 RELATED WORK

**Causal representation learning**  Our work builds on the nascent field of causal representation learning (Schölkopf et al., 2021). In particular, our disentanglement approaches builds on ideas in Ahuja et al. (2022b) which uses the same assumptions as Locatello et al. (2020a) but relaxes the requirement that the latent variables are independently distributed. These approaches form part of a larger body of recent work that shows the importance of sparsity and weak supervision from actions in disentanglement (Lachapelle et al., 2022; Lachapelle & Lacoste-Julien, 2022; Brehmer et al., 2022; Lippe et al., 2022; 2023b;a). In the appendix, we also show how known mechanisms from Ahuja et al. (2022a) can be dealt with in our framework. A closely related, but more general setting, is the recent progress on disentanglement from interventional distributions which do not require paired samples (Ahuja et al., 2023; Buchholz et al., 2023; von Kügelgen et al., 2023); we believe a useful extension of our approach would consider these settings. This literature builds on the foundational work from the nonlinear independent component analysis (ICA) literature (Hyvarinen & Morioka, 2016; 2017; Hyvarinen et al., 2019; Khemakhem et al., 2020a).

**Object-centric learning.**  Natural data can often be decomposed into smaller entities—objects—that explain the data. The overarching goal of object-centric learning is to model such data in terms of these multiple objects. The reason for this is simple: it is usually easier to reason over a small set of relevant objects rather than, for example, a large grid of feature vectors. Representing data in this way has downstream benefits like better robustness (Huang et al., 2020). An important line of research in this area is how to obtain such objects from data like images and video in the first place. Typically, a reconstruction setup is used: given an image input, the model learns the objects in the latent space, which are then decoded back into the original image with a standard reconstruction loss (Locatello et al., 2020b; van Steenkiste et al., 2018b). Nguyen et al. (2023) propose RSM, a conceptually close idea to our work. They jointly learn object-centric representations with a modular dynamics model by minimizing a rolled out reconstruction loss. However, they do not obtain any disentanglement of object properties, and the form of our proposed weak-supervision provides insights to the effectiveness of their method for improving generalization.

We use slot attention since it makes very few assumptions about the desired data. For instance, some methods model foreground differently from background. Additionally, DINOSAUR (Seitzer et al., 2022) shows recent success on more complex images, which demonstrates the versatility of the slot attention approach. While in general object-centric models operate on image inputs and thus identify visual objects, it is in principle applicable to other domains like audio (Reddy et al., 2023) as well.

## 7 EMPIRICAL EVALUATION

**Setup.**  We evaluated our method on 2D and 3D synthetic image datasets that allowed us to carefully control various aspects of the environment, such as the number of objects, their sizes, shapes, colors, relative position, and dynamics. Examples of our 2D and 3D datasets are shown in figures 1,3 respectively. The object-wise true latents in either dataset consist of $z = (p_x, p_y, h, s, r, \phi)$, where $p_x, p_y$ denote the coordinates of the center of an object, followed by color hue $h$, shape $s$, size $r$, and rotation angle $\phi$ about the z-axis. Therefore, we deal with both discrete and continuous properties. For further details on the dataset generation, see appendix F.6.

**Disentanglement Metrics.**  We compared $\hat{z}$—the projections of non-background slots—to the true latents $z$ of objects to measure the disentanglement of the properties in $\hat{z}$. We evaluated identifiability of the learned representations either up to affine transformations or up to permutation and scaling. These two metrics were computed by fitting a linear regression between $z, \hat{z}$ and reporting the

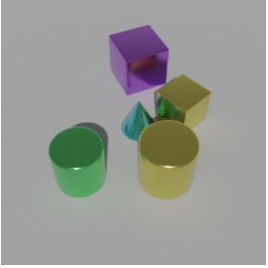 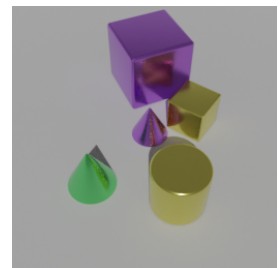

Figure 3: *(Left)* An example image before any perturbation. *(Right)* Possible perturbations in the synthetic 3D dataset, i.e. change in size, orientation, color, position, and shape.

coefficient of determination $R^2$, and using the mean correlation coefficient (MCC) (Hyvarinen & Morioka, 2016; 2017).

Table 1: Permutation Disentanglement (MCC) scores on 2D shapes test set under *unknown* fully sparse perturbations. All results are averaged over 3 seeds except those requiring to train SA-MESH from scratch that were trained only once. SA-LR (supervised) achieves a score of 1.0 in all settings.

| Model | $\mathrm{pos}_x, \mathrm{pos}_y$ | | | $\mathrm{pos}_x, \mathrm{pos}_y, \mathrm{color}, \mathrm{size}, \mathrm{rotation}$ | | |
|---|---|---|---|---|---|---|
| | $n=2$ | $n=3$ | $n=4$ | $n=2$ | $n=3$ | $n=4$ |
| Ours | **1.00** ±0.01 | **1.00** ±0.01 | 0.98 ±0.01 | **0.95** ±0.01 | **0.93** ±0.00 | **0.94** ±0.01 |
| SA-RP | 0.80 | 0.90 | 0.82 | 0.58 | 0.52 | 0.50 |
| SA-PC | **1.00** | **1.00** | **1.00** | 0.86 | 0.85 | 0.84 |
| CNN[†] | 0.96 ±0.02 | 0.99 ±0.01 | 0.98 ±0.02 | 0.91 ±0.01 | 0.89 ±0.01 | 0.90 ±0.01 |
| CNN | 0.40 ±0.01 | 0.25 ±0.03 | 0.21 ±0.01 | 0.58 ±0.00 | 0.42 ±0.00 | 0.27 ±0.01 |

Table 2: LD scores on 3D shapes test set under *unknown* fully sparse perturbations. SA-LR achieves a score of 1.0 in all settings.

| Model | $\mathrm{pos}_x, \mathrm{pos}_y, \mathrm{color}$ | | | $\mathrm{pos}_x, \mathrm{pos}_y, \mathrm{color}, \mathrm{size}, \mathrm{rotation}$ | | |
|---|---|---|---|---|---|---|
| | $n=2$ | $n=3$ | $n=4$ | $n=2$ | $n=3$ | $n=4$ |
| Ours | **0.99** ±0.01 | **0.99** ±0.00 | **1.00** ±0.01 | **0.91** ±0.03 | **0.95** ±0.01 | **0.93** ±0.01 |
| SA-RP | 0.67 | 0.58 | 0.58 | 0.51 | 0.56 | 0.60 |
| SA-PC | 0.64 | 0.62 | 0.64 | 0.56 | 0.76 | 0.76 |

**Baselines.** Our baselines were selected to use a series of linear probes to evaluate how much of the property information is already present in the slot representations $s_i$ of a vanilla SA-MESH implementation which was only trained for reconstruction. We compare this with our approach which explicitly optimizes to disentangle the properties with the weakly supervised loss in Eq 2. For the baselines, we mapped from the slot representations to a $d$-dimensional latent space with *random projections* (RP)—which preserve distances in the projected space to obtain a crude estimate of vanilla slot attention disentanglement—the first $d$ *principal components* (PC), and *linear regression* (LR) which provides a supervised upper-bound on what is achievable with linear maps. Finally, we also included a standard ResNet18 (He et al., 2016) (denoted *CNN*) trained with Ahuja et al. (2022b)'s procedure that does not address injectivity issues, and the same trained on a DGP which is modified to be injective[6] (denoted *CNN[†]*).

**2D Shapes.** The results in Table 1 (with additional results in appendix E.1) confirmed that as long as the generative function is injective we can empirically achieve identification (see *CNN[†]*). But

---

[6]We make $g$ injective by using the properties that are not the target of disentanglement, i.e., if $x, y$ are the target properties, we will uniquely color each object based on its order in the default permutation. If $x, y, c$ are targets, we will use unique shapes for each object based on its order in the permutation. The same logic follows for other property combinations.

Table 3: MCC scores on 3D shapes test set under *unknown* fully sparse perturbations. SA-LR achieves a score of 1.0 in all settings.

| | $\text{pos}_x, \text{pos}_y, \text{color}$ | | | $\text{pos}_x, \text{pos}_y, \text{color}, \text{size}, \text{rotation}$ | | |
|---|---|---|---|---|---|---|
| Model | $n = 2$ | $n = 3$ | $n = 4$ | $n = 2$ | $n = 3$ | $n = 4$ |
| Ours | **0.99** $\pm 0.01$ | **0.99** $\pm 0.00$ | **0.99** $\pm 0.01$ | **0.89** $\pm 0.02$ | **0.92** $\pm 0.03$ | **0.92** $\pm 0.02$ |
| SA-RP | 0.62 | 0.54 | 0.54 | 0.49 | 0.50 | 0.46 |
| SA-PC | 0.69 | 0.68 | 0.70 | 0.64 | 0.77 | 0.78 |

the moment we drop any ordering over the objects and render $x$ via a non-injective function, then identification via ResNet18, which is suited only to injective generative functions, fails disastrously (see the row corresponding to CNN in table 4. On the other hand, we can see that our method has no difficulty identifying object properties because it treats them as a set by leveraging slot attention and a matching procedure. Additionally, the shared structure of learned latents in our method significantly improves the sample efficiency for disentanglement (see appendix E.3). The strong performance of the principal components of vanilla SA-MESH on the position disentanglement task likely results from the positional encoding. On the more complex task that also involves color, size and rotation, MCC performance drops for SA-PC, though it is still surprisingly high given that the model is just trained for reconstruction. This is likely because these are very simple images with properties that were selected independently, uniformly at random so the slot principal components align with the ground-truth axes of variation in the data.

**3D Shapes.** Figure 3 shows examples of perturbations that the model observes and uses for disentangling object properties in tables 2 and 3. We present the disentanglement scores for various combinations of properties and environments with $k = \{2, 3, 4\}$ number of objects in the scene. Since non-injective CNN failed consistently in the simpler 2D dataset, we do not evaluate it with 3D shapes. Results for our method are averaged over 3 seeds, but since the baselines require training SA-MESH from scratch, they were trained only once as it is computationally expensive to obtain excellent reconstructions with slot attention. These results essentially confirm our findings in the simpler 2D dataset, and demonstrate how treating the scene as a set with our method results in perfect disentanglement of object properties. For the results on other combinations of properties, please see appendix E.2.

## 8 CONCLUSION

This study establishes a connection between causal representation learning and object-centric learning, and (to the best of our knowledge) for the first time shows how to achieve disentangled representations in environments with multiple interchangeable objects. The importance of recognizing this synergy is two-fold. Firstly, causal representation learning has largely ignored the subtleties of objects in assuming injectivity and fixed $\mathbb{R}^d$ representations. Conversely, object-centric learning has not dealt with the challenge of unsupervised disentanglement. Yet disentangled representations can significantly improve a model's generalization capabilities under distribution shifts, and could also allow for learning parsimonious models of the dynamics when such proper representations are achieved, which we deem as important avenues for future research. In this study we provided empirical evidence showcasing the successful disentanglement of object-centric representations through the fusion of slot attention with recent advances in causal representation learning.

## 9 ACKNOWLEDGMENTS

The authors would like to thank Kartik Ahuja for the helpful discussions and feedback on the initial directions of this work. This research was enabled by compute resources and technical support provided by Mila, Quebec AI Institute (mila.quebec) as well as the Digital Research Alliance of Canada, formerly known as Compute Canada. Amin Mansouri was supported in part by the support from Canada CIFAR AI Chair Program, from the Canada Excellence Research Chairs Program, and Microsoft.

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

## A  PROOF OF THEOREM 1

We want to compare the number of perturbations needed to disentangle shared properties with a standard encoder to those needed by an object-centric encoder. Our strategy will be as follows,

1. Setup a data generating process with multiple objects where injectivity holds by construction so that we can restate Theorem 1 from Ahuja et al. (2022b) to show they need $k \times d$ perturbations.

2. Define an object-centric architecture in terms of the object-wise partitions that we defined in Definition 1.

3. Restate an analog of Theorem 1 from Ahuja et al. based on the object-centric encoder.

4. Theorem 1 in the main text will follow as a collary of the difference between the number of perturbations used in the two theorems above.

We begin by defining a data generating process such that $g(\mathbf{vec}_\pi(Z))$ is injective by construction. We can achieve this by appending an id, $i$, to each $z_i$ in $Z$, such that $Z = \{z_i \oplus [i]\}_{i=1}^k$ where $\oplus$ denotes concatenation, and then choosing $\mathfrak{g}$ such that $x$ depends on $i$ (for example, each $i$ could be rendered in a different color). Like Ahuja et al., we assume we have data that is perturbed by $\Delta := \{\{\delta_{i,j}\}_{j=1}^d\}_{i=1}^k$, a set of 1-sparse perturbations that perturbs each of the $d$ properties from each of the $k$ objects. Taken together, we have the following data generating process (DGP),

$$Z = \{z_i \oplus [i]\}_{i=1}^k \sim \mathbb{P}_Z, \; x := \mathfrak{g}(Z) \quad \tilde{z}_{j,l} := z_j + \delta_{j,l} \, \forall \delta_{i,j} \in \Delta, \quad \tilde{x}_{j,l} := \mathfrak{g}(\{z_1, \dots, \tilde{z}_{j,l}, \dots, z_k\}) \tag{3}$$

where each object has $d$ shared properties, $z_i \in \mathbb{R}^d$, and $z_{i,j}$ and $z_{i',j}$ are of the same type—e.g. position $x$, hue, etc.— for all $j$. As before, assume $\mathfrak{g}$ is injective, and define $g^* = g(\mathbf{vec}_{\pi^*}(Z))$ where $\pi^*$ is the permutation that sorts $Z$ by the index $i$, so that $g^*$ is injective by construction.

Now, Ahuja et al. show that if the encoder, $\hat{f} : \mathcal{X} \to \mathbb{R}^{kd}$, is chosen to minimize the following loss,

$$\hat{f} \in \arg\min_{f'} E_{x,x',\delta} \left[ (f'(x) + \delta - f'(x'))^2 \right] \tag{4}$$

and the following assumptions hold,

**Assumption 1.** *The dimension of the span of the perturbations in* equation 3 *is* $kd$, *i.e.,* $\mathsf{dim}\Big(\mathsf{span}(\Delta)\Big) = kd$.

**Assumption 2.** $a(z) := f \circ g^*(z)$ *is an analytic function. For each component* $i \in \{1, \cdots, kd\}$ *of* $a(z)$ *and each component* $j \in \{1, \cdots, kd\}$ *of* $z$, *define the set* $\mathcal{S}^{ij} = \{\theta \mid \nabla_j a_i(z + b) = \nabla_j a_i(z) + \nabla_j^2 a_i(\theta)b, z \in \mathbb{R}^{kd}\}$, *where* $b$ *is a fixed vector in* $\mathbb{R}^{kd}$. *Each set* $\mathcal{S}^{ij}$ *has a non-zero Lebesgue measure in* $\mathbb{R}^{kd}$.

Then we have,

**Theorem 2** ((Ahuja et al., 2022b)). *Assume we have data from the DGP in equation 3 and assumption 1 and 2 hold and the number of perturbations per example equals the latent dimension,* $m = kd$, *then the encoder that solves equation 3 identifies true latents up to permutation and scaling, i.e.* $\hat{z} = \Pi\Lambda z + c$, *where* $\Lambda \in R^{kd \times kd}$ *is an invertible diagonal matrix,* $\Pi \in R^{kd \times kd}$ *is a permutation matrix and* $c$ *is an offset.*

*Proof.* See Ahuja et al. for the proof. $\qquad\square$

Now, consider an object-centric architecture encoder of the form $F(x) := \{\mathfrak{f}(x^i)\}_{x^i \in P}$ where $P$ is an object-wise partition and $\mathfrak{f} : \mathcal{X} \to \mathbb{R}^d$. Let $\sigma \in \Sigma$ denote a permutation of the latents from the set of all $k-$permutations. Let:

$$\hat{F}(x) \in \arg\min_{\mathfrak{f}} E_{x,x',\delta}[\min_{\sigma' \in \Sigma} \|((\mathfrak{f}(x')^{\sigma'(i)}) - ((\mathfrak{f}(x)^{(i)}) + \delta)\|^2] \qquad (5)$$

Note that since $\delta$ is non-zero for only one pair of patches $x^{(i)}, x^{(i)'}$ and zero otherwise, the minimizer over $\Sigma$ is almost surely unique. Assumptions 3 and 4 are analogs of 1 and 2 above, but make reference to the dimensionality of the co-domain of $\mathfrak{f}$ rather than $f$.

**Assumption 3.** *The dimension of the span of the perturbations in* equation 3 *is* $d$, *i.e.,* $\mathsf{dim}\Big(\mathsf{span}(\Delta)\Big) = d$.

**Assumption 4.** $a(z) := \mathfrak{f} \circ g^*(z)$ *is an analytic function. For each component* $i \in \{1, \cdots, d\}$ *of* $a(z)$ *and each component* $j \in \{1, \cdots, d\}$ *of* $z$, *define the set* $\mathcal{S}^{ij} = \{\theta \mid \nabla_j a_i(z + b) = \nabla_j a_i(z) + \nabla_j^2 a_i(\theta)b, z \in \mathbb{R}^d\}$, *where* $b$ *is a fixed vector in* $\mathbb{R}^d$. *Each set* $\mathcal{S}^{ij}$ *has a non-zero Lebesgue measure in* $\mathbb{R}^d$.

With this setup, the following theorem follows directly from Theorem 2 as a reduction from the multi-object to single-object setting,

**Theorem 3.** *Assume we have data from the DGP in equation 3 and assumption 3 and 4 hold and the number of perturbations per example equals the latent dimension,* $m = d$, *then the encoder that solves equation 5 for an object-wise partition P, identifies true latents up to permutation and scaling, i.e.* $\hat{z} = \Pi\Lambda z + c$, *where* $\Lambda \in R^{kd \times kd}$ *is an invertible diagonal matrix,* $\Pi \in R^{kd \times kd}$ *is a permutation matrix and* $c$ *is an offset.*

*Proof.* Because $P$ is an object-wise partition, the function that produces each $x^{(i)} \in P$ is injective with respect to some $z_{\sigma(i)}$ (i.e. one of the object's latents). Thus for each $x^{(i)}$, the solution to equation 5 is equivalent to the single object setting with $k = 1$, and thus theorem 2 applies, which implies that $\mathfrak{f}(x^{(i)}) = \hat{z}_i = \Pi\Lambda z_i + c$ for all $i$. Now let $\hat{z} = \mathbf{vec}_\pi(\{\hat{z}_i\}_{i=1}^k)$. Because each $\hat{z}_i$ is identified up to a permutation, scaling and offset, and for any $\pi$, there exists a $\Pi$ such that $\hat{z} = \Pi\Lambda z + c$ which completes the result. $\qquad\square$

**Corollary 1.** *If the assumptions for Theorem 2 and 3 hold and a data generating process outputs observations containing $k$ objects with shared properties, then an object-centric architecture of the form $F(x) := \{\mathfrak{f}(x^{(i)})\}_{x^{(i)} \in P}$ will disentangle in $1/k$ fewer perturbations than an encoder of the form $\mathfrak{f} : \mathcal{X} \to \mathbb{R}^{kd}$.*

*Proof.* This follows directly from comparing the the number of perturbations required in Theorems 2 and 3. $\qquad\square$

## B    BACKGROUND ON SLOT-ATTENTION-BASED ARCHITECTURES

Slot attention (Locatello et al., 2020b) is a neural network component that, intuitively, summarizes the relevant information in the input set (most commonly, image features with position embeddings) into a smaller set of so-called "slots". Each slot is a feature vector that can be thought of as capturing information about one "object" in the input set, which usually comprises multiple elements of the input set. This is done by repeating cross-attention between the inputs and the slots to compute per-slot updates.

In the traditional set-up, these slots are then used to reconstruct the input with an auto-encoder objective: each slot is decoded into a separate image through a shared image decoder, which is followed by merging these per-slot images into a single reconstructed image. Ideally, slot attention is able to decompose the original image into distinct objects, each of which is modeled by a single slot.

More concretely, slot attention takes as input a matrix $\mathbf{X} \in \mathbb{R}^{n \times c}$ with $n$ as the number of inputs and $c$ the dimensionality of each input. We also randomly initialize the slots $\mathbf{Z}^{(0)} \in \mathbb{R}^{m \times d}$. We start by computing the query, key, and value matrices as part of cross-attention.

$$\mathbf{Q}^{(t)} = \mathbf{Z}^{(t)} \mathbf{W}_Q \quad \mathbf{K} = \mathbf{X} \mathbf{W}_K \quad \mathbf{V} = \mathbf{X} \mathbf{W}_V \tag{6}$$

This is followed by the normalized cross-attention to determine the attention map $\mathbf{A} \in \mathbb{R}^{m \times n}$, then a GRU (Cho et al., 2014) to apply this update to the slots.

$$\mathbf{A}^{(t)} = \mathrm{normalize}(\mathbf{Q}^{(t)} \mathbf{K}^\top) \tag{7}$$

$$\mathbf{Z}^{(t+1)} = \mathrm{GRU}(\mathbf{Z}^{(t)}, \mathbf{A}^{(t)} \mathbf{V}) \tag{8}$$

The function $\mathrm{normalize}$ encourages slots to compete for inputs by applying a softmax over slots and normalizing the weights for each input to sum to one. After $T$ steps, the algorithm outputs $\mathbf{Z}^{(T)}$, a set of $m$ embedding vectors $\{\mathbf{z}_i^{(T)}\}_{i=1}^m$ that can be used as input to a shared image decoder.

**SA-MESH.**    The specific version of slot attention we use in this paper is SA-MESH (Zhang et al., 2023). It makes regular slot attention more powerful by giving it the ability to break ties between slots more effectively. [7] In practice, this improves the quality of the individual slot representations significantly due to less mixing of unrelated inputs into the same slot.

The key difference with regular slot attention is that it features an entropy minimization procedure to approximate an optimal transport solution, which makes the attention map sparse. The connection to optimal transport is made by the use of the standard Sinkhorn algorithm (Sinkhorn & Knopp, 1967; Cuturi, 2013).

$$\mathrm{MESH}(\mathbf{C}) = \arg\min_{\mathbf{C}} H(\mathrm{sinkhorn}(\mathbf{C})) \tag{9}$$

$$\mathbf{A}^{(t)} = \mathrm{sinkhorn}(\mathrm{MESH}(\mathbf{Q}^{(t)} \mathbf{K}^\top)) \tag{10}$$

The optimization problem is solved by unrolling gradient descent, with a noisy initialization to ensure that ties are broken.

## C    ALTERNATIVE PERTURBATION MECHANISMS

**Dense vs. Sparse.**    We can have a number of assumptions on the perturbation mechanisms and the nature of model's knowledge about those mechanisms. In the most general case, suppose $\mathcal{M} = \{m^1(\cdot), m^2(\cdot), ..., m^k(\cdot)\}$ denotes the set of all possible perturbation mechanisms. To obtain $x'$, the perturbed variant of $x$, we then select a subset of $k' \leq k$ objects as targets that undergo perturbations determined by a subset of $k'$ mechanisms $\mathcal{M}' \subset \mathcal{M}$ from the set of all possible perturbation mechanisms. The correspondence between the $k'$ mechanisms and $k'$ perturbed objects is decided by a random permutation $\pi_t^{\mathcal{M}}$, i.e. $i = \pi_t^{\mathcal{M}}[j]$ means that mechanism $i$ governs the transition dynamics of object $j$ to produce $z_{t+1}^j$ (for objects that are not supposed to change from

---

[7]Concretely, it makes the mapping from the initial slots to the final slots *exclusively multiset-equivariant* (Zhang et al., 2022) rather than permutation-equivariant/set-equivariant.

$t \to t+1$ a dummy mechanism with index $-1$ can be assumed which results in no change). A mechanism $m^i(\cdot) : \mathbb{R}^d \to \mathbb{R}^d$ in $\mathcal{M}$ is a vector-valued function that operates on object-wise true latents $z_t^j$ and outputs $z_{t+1}^j = z_t^j + \delta_i$ such that $i = \pi_t^{\mathcal{M}}[j]$. Perturbation vectors $\delta_i$ could be sparse or not. The subset $\mathcal{M}'$ can contain $k' = k$ mechanisms to perturb all of the $k$ objects in the environment, and if none of the $k' = k$ resulting perturbations is sparse, we denote the set $\mathcal{M}'$ as *fully dense perturbations*, i.e., all of the properties of all objects will change from $t \to t+1$. $\mathcal{M}'$ can also contain at least one object but not all of them ($1 \le k' < k$) with sparse or dense perturbations, or it may consist of only a single object ($k' = 1$) that is perturbed by a fully sparse mechanism, one that only alters a single property and leaves the rest unchanged. We denote this scenario as *fully sparse perturbation*.

## D  MATCHING

Perturbations alter the properties of objects from $t \to t+1$ and the model has to infer which object's properties were perturbed to update its representations and minimize the latent loss equation 1. But recall that the model has no direct access to objects. It receives the observations at $t, t+1$ and encodes each of them to a set of slots $\mathcal{S}_t, \mathcal{S}_{t+1}$. These slots do not follow any fixed ordering, and moreover, there is no guarantee that each slot binds to exactly one unique object. Slots can also correspond to the background. Each perturbation $\delta^i$ changes the properties of some object $z_t^j$, so the model requires to find a pair of slots $(s_t^u, s_{t+1}^v)$ that are bound to object $z^j$ at $t$ and $t+1$, respectively. Once the model figures out such a matching, then the latent loss that results in disentanglement can be computed via the projections of these slots $\hat{z}$:

$$\mathcal{L}_z = \sum_{i=1}^m \| \hat{z}_t^i + \delta_t^{\pi_t^{\mathcal{M}'}[i]} - \hat{z}_{t+1}^i \|^2, \qquad \hat{z}_t^i = p(s_t^u), \;\; \hat{z}_{t+1}^i = p(s_t^v) \qquad (11)$$

The problem of finding a correspondence between slot projections at $t, t+1$ and the perturbations is an instance of the 3-dimensional matching. However, recall that we use fully sparse perturbations for our mechanisms, thus, the problem is significantly simplified.

When the model is presented with fully sparse perturbations, the 3-dimensional matching reduces to finding the minimum element in a $1-d$ cost array of size $|\mathcal{S}|^2$, where $|\mathcal{S}|$ is the number of slots. The reason is that since only one property of a single object $z_t^i$ is being altered (and the model knows perturbations are fully sparse), the model can apply that 1-sparse perturbation $\delta_t$ (or the transformed version in the unknown setting) to all of slot projections $\hat{z}_t^j$ for $j \in \{1, \ldots, |\mathcal{S}|\}$ at time $t$, and only find *one pair* of slots from the set of $|\mathcal{S}|^2$ possible pairs at $t, t+1$, which correspond to the perturbed object:

$$(i, j) = \arg\min_{i', j'} \| \hat{z}_{t+1}^{j'} - (\hat{z}_t^{i'} + \delta_t) \|^2 \qquad (12)$$

However, this minimization can be made simpler if we use the slots obtained at $t$ to initialize the slots at $t+1$. This way, in practice, the order of slots at $t, t+1$ is very likely to be preserved, not only since the slots at $t+1$ are initialized with hints from $t$, but also the sparse perturbation makes it much easier for all slots to bind to the same object as only a small subset of the scene needs to be readjusted among the slots. Therefore the matching would reduce to a simple minimization over $|\mathcal{S}|$ elements:

$$i = \arg\min_{i'} \| \hat{z}_{t+1}^{i'} - (\hat{z}_t^{i'} + \delta_t) \|^2 \qquad (13)$$

Note however that we still would use all slot projections for evaluation, the only difference with this matching scheme is that gradient signals are only propagated from the perturbed object (as they also should, since there is no change in other slots, there is nothing there to be learned that could help disentanglement.)

## E  FURTHER EXPERIMENTAL RESULTS

### E.1  2D SHAPES

Table 4 gives the linear disentanglement scores for the same experiment as that shown in table 1 in the main text.

Table 4: Linear Disentanglement (LD) scores on 2D shapes test set under *unknown* fully sparse perturbations. All results are averaged over 3 seeds except those requiring to train SA-MESH from scratch that were trained only once. SA-LR, which is supervised by the ground truth latents, and is an upper bound on the disentanglement performance, achieves a score of 1.0 in all settings.

| Model | $\text{pos}_x, \text{pos}_y$ | | | $\text{pos}_x, \text{pos}_y, \text{color}, \text{size}, \text{rotation}$ | | |
|---|---|---|---|---|---|---|
| | $n=2$ | $n=3$ | $n=4$ | $n=2$ | $n=3$ | $n=4$ |
| Ours | **1.00** $\pm0.00$ | **1.00** $\pm0.00$ | **1.00** $\pm0.00$ | **0.95** $\pm0.01$ | **0.93** $\pm0.01$ | **0.94** $\pm0.02$ |
| SA-RP | 0.92 | 0.96 | 0.94 | 0.75 | 0.70 | 0.68 |
| SA-PC | 1.00 | 1.00 | 1.00 | 0.93 | 0.88 | 0.86 |
| CNN$^\dagger$ | 0.94 $\pm0.05$ | 0.99 $\pm0.00$ | 0.96 $\pm0.03$ | 0.87 $\pm0.01$ | 0.84 $\pm0.01$ | 0.86 $\pm0.01$ |
| CNN | 0.24 $\pm0.01$ | 0.13 $\pm0.01$ | 0.07 $\pm0.01$ | 0.35 $\pm0.00$ | 0.19 $\pm0.00$ | 0.08 $\pm0.01$ |

Tables 5,6 extend our results under unknown fully sparse perturbations on the 2D shapes dataset to more combinations of disentanglement target properties. We can observe that our results stay very close to the upper bound on the achievable performance which uses a *supervised* linear regression from slot projections $\hat{z}$ to *ground truth* latents $z$. These tables highlight once again the pivotal role of the injectivity assumption in achieving identification with conventional encoders that ignore the object-centricity of the environment (see the performance drop from *CNN*$^\dagger$ to CNN, where the latter drops the unrealistic injectivity assumption).

Table 5: Linear Disentanglement (LD) scores on 2D shapes test set under *unknown* fully sparse perturbations. All results are averaged over 3 seeds except those requiring to train SA-MESH from scratch that were trained only once. SA-LR achieves a score of 1.0 in all settings.

| Model | $\text{pos}_x, \text{pos}_y, \text{color}$ | | | $\text{pos}_x, \text{pos}_y, \text{shape}$ | | |
|---|---|---|---|---|---|---|
| | $n=2$ | $n=3$ | $n=4$ | $n=2$ | $n=3$ | $n=4$ |
| Ours | **1.00** $\pm0.01$ | **0.98** $\pm0.01$ | **0.99** $\pm0.00$ | **1.00** $\pm0.01$ | **0.98** $\pm0.01$ | **0.99** $\pm0.01$ |
| SA-RP | 0.77 | 0.61 | 0.60 | 0.71 | 0.68 | 0.70 |
| SA-PC | 0.97 | 0.98 | 0.99 | 0.80 | 0.66 | 0.87 |
| CNN$^\dagger$ | 1.00 $\pm0.00$ | 0.99 $\pm0.01$ | 0.98 $\pm0.00$ | 1.00 $\pm0.00$ | 1.00 $\pm0.00$ | 0.99 $\pm0.00$ |
| CNN | 0.35 $\pm0.00$ | 0.15 $\pm0.00$ | 0.07 $\pm0.01$ | 0.32 $\pm0.01$ | 0.15 $\pm0.01$ | 0.11 $\pm0.01$ |

| Model | $\text{pos}_x, \text{pos}_y, \text{color}, \text{shape}$ | | |
|---|---|---|---|
| | $n=2$ | $n=3$ | $n=4$ |
| Ours | **0.99** $\pm0.00$ | **0.98** $\pm0.01$ | **0.99** $\pm0.00$ |
| SA-RP | 0.69 | 0.73 | 0.60 |
| SA-PC | 0.74 | 0.75 | 0.52 |
| CNN$^\dagger$ | 1.00 $\pm0.00$ | 0.99 $\pm0.01$ | 1.00 $\pm0.00$ |
| CNN | 0.40 $\pm0.00$ | 0.21 $\pm0.00$ | 0.11 $\pm0.00$ |

## E.2 3D SHAPES

**Quantitative Results.** Tables 7, 8 extend our results under unknown fully sparse perturbations on the 3D shapes dataset to more combinations of disentanglement target properties. Again, we can observe the applicability of our method to this more complex 3D dataset that contains artifacts related to depth, occlusion, and lighting, to name a few. Again our results stay very close to the upper bound on the achievable performance which uses a *supervised* linear regression from slot projections $\hat{z}$ to *ground truth* latents $z$.

**Qualitative Results.** Figures 4-6 illustrate the learned disentangled (object-centric) representations. Each figure shows a sequence of 3D samples evolving over 5 steps (shown on the left), and how the learned representations respond to the perturbations (shown on the right). Perturbations include changing the object's $\text{pos}_x, \text{pos}_y$, color, $\phi$ (rotation). The model used here is trained with 3 slots, and the learned representations are the result of a projection layer learned via our weakly-supervised method applied to $64-$dimensional object-centric representations. The projection maps each slot

Table 6: Permutation Disentanglement (MCC) scores on 2D shapes test set under *unknown* fully sparse perturbations. All results are averaged over 3 seeds except those requiring to train SA-MESH from scratch that were trained only once. SA-LR achieves a score of 1.0 in all settings.

| Model | $pos_x, pos_y, color$ | | | $pos_x, pos_y, shape$ | | |
|---|---|---|---|---|---|---|
| | $n = 2$ | $n = 3$ | $n = 4$ | $n = 2$ | $n = 3$ | $n = 4$ |
| Ours | **1.00** $\pm 0.01$ | **0.95** $\pm 0.05$ | **0.97** $\pm 0.02$ | **0.99** $\pm 0.01$ | **0.99** $\pm 0.01$ | **0.99** $\pm 0.01$ |
| SA-RP | 0.74 | 0.60 | 0.60 | 0.66 | 0.63 | 0.59 |
| SA-PC | 0.87 | 0.89 | 0.90 | 0.83 | 0.81 | 0.89 |
| CNN[†] | 1.00 $\pm 0.00$ | 0.99 $\pm 0.01$ | 0.99 $\pm 0.01$ | 1.00 $\pm 0.00$ | 1.00 $\pm 0.00$ | 0.99 $\pm 0.00$ |
| CNN | 0.55 $\pm 0.01$ | 0.35 $\pm 0.01$ | 0.24 $\pm 0.01$ | 0.52 $\pm 0.02$ | 0.33 $\pm 0.02$ | 0.28 $\pm 0.02$ |

| Model | $pos_x, pos_y, color, shape$ | | |
|---|---|---|---|
| | $n = 2$ | $n = 3$ | $n = 4$ |
| Ours | **0.99** $\pm 0.01$ | **0.98** $\pm 0.01$ | **0.99** $\pm 0.01$ |
| SA-RP | 0.54 | 0.68 | 0.55 |
| SA-PC | 0.64 | 0.63 | 0.57 |
| CNN[†] | 1.00 $\pm 0.00$ | 0.99 $\pm 0.01$ | 1.00 $\pm 0.00$ |
| CNN | 0.61 $\pm 0.00$ | 0.43 $\pm 0.00$ | 0.30 $\pm 0.00$ |

Table 7: Linear Disentanglement (LD) scores on 3D shapes test set under *unknown* fully sparse perturbations. All results are averaged over 3 seeds except those requiring to train SA-MESH from scratch that were trained only once. SA-LR achieves a score of 1.0 in all settings.

| Model | $pos_x, pos_y, size$ | | | $pos_x, pos_y, color, rotation$ | | |
|---|---|---|---|---|---|---|
| | $n = 2$ | $n = 3$ | $n = 4$ | $n = 2$ | $n = 3$ | $n = 4$ |
| Ours | **0.98** $\pm 0.01$ | **0.98** $\pm 0.01$ | **0.98** $\pm 0.00$ | **0.98** $\pm 0.00$ | **0.97** $\pm 0.01$ | **0.98** $\pm 0.01$ |
| SA-RP | 0.61 | 0.62 | 0.53 | 0.59 | 0.54 | 0.55 |
| SA-PC | 0.78 | 0.84 | 0.78 | 0.70 | 0.72 | 0.69 |

Table 8: Permutation Disentanglement (MCC) scores on 3D shapes test set under *unknown* fully sparse perturbations. All results are averaged over 3 seeds except those requiring to train SA-MESH from scratch that were trained only once. SA-LR achieves a score of 1.0 in all settings.

| Model | $pos_x, pos_y, size$ | | | $pos_x, pos_y, color, rotation$ | | |
|---|---|---|---|---|---|---|
| | $n = 2$ | $n = 3$ | $n = 4$ | $n = 2$ | $n = 3$ | $n = 4$ |
| Ours | **0.96** $\pm 0.02$ | **0.96** $\pm 0.05$ | **0.96** $\pm 0.03$ | **0.98** $\pm 0.01$ | **0.98** $\pm 0.02$ | **0.97** $\pm 0.01$ |
| SA-RP | 0.60 | 0.57 | 0.52 | 0.55 | 0.51 | 0.49 |
| SA-PC | 0.87 | 0.90 | 0.86 | 0.73 | 0.76 | 0.74 |

from $\mathbb{R}^{64} \to \mathbb{R}^4$, i.e., the disentanglement target space. In figures 4-6, the 4 dimensions of such projections for object slots are presented over 5 steps, i.e., each set of colored lines shows the evolution of the projection of a slot corresponding to the object with the same color. Please refer to the figures for details of the perturbations. Lastly, we have kept the number of objects in these scenes to two for clarity of the presentation, however tables 2,3, 7,8 show that we achieve similar performances with other sets of properties and number of objects in the scene.

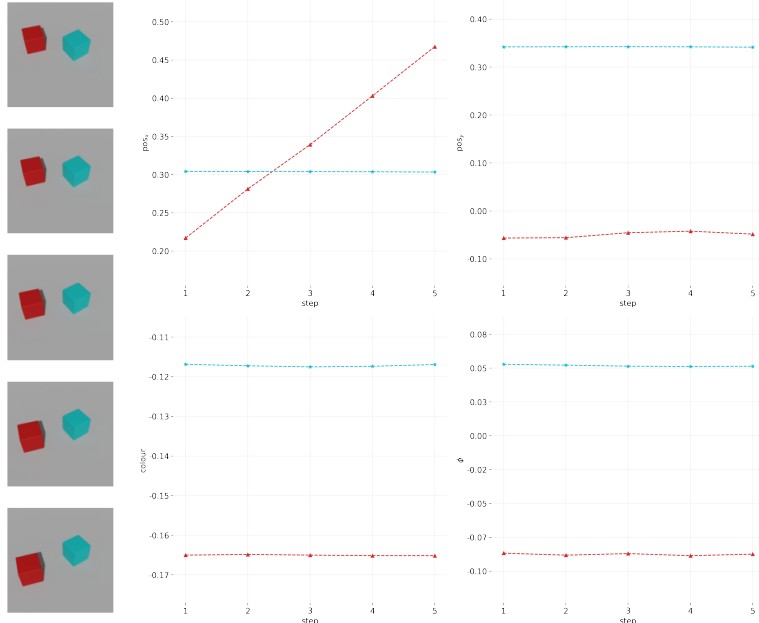

Figure 4: *(Left)* From top to bottom, each steps perturbs the $\text{pos}_x$ coordinate of the red object by $0.2$ in the *ground-truth latent space*, *(Right)* which is reflected (through an affine transform) as a linear increase in $\text{pos}_x$, while the rest of the properties for both objects remain the same, demonstrating that the learned representations are indeed disentangled. In generating these samples, camera has some non-zero angle w.r.t. the origin, therefore, perturbations in the $x$ direction appear as vertical displacements.

### E.3 COMPARISON OF SAMPLE EFFICIENCY

Figure 7 demonstrates the sample efficiency of our object-centric model compared to a ResNet that achieves disentanglement with an injective DGP. Both models are trained with varying number of training samples that contain $k = 4$ objects for which $\text{pos}_x, \text{pos}_y$, color, shape are the disentanglement target properties. Since we sample 1-sparse perturbations uniformly, the training dataset size could be thought of as a proxy for the number of different perturbations a given configuration of objects would encounter. Although according to theory, the injective ResNet should require *at least $k$* times more perturbations to identify the latents up to affine transformations, we observe that the advantage of our object-centric model in terms of sample efficiency is much more pronounced in practice. Our method can achieve close to perfect disentanglement with as few as 100 training samples, while an injective ResNet takes 100 times more samples to raise to a comparable performance. This highlights the practical importance of exploiting the inherent set structure of objects in a scene for representation learning.

## F IMPLEMENTATION AND EXPERIMENTAL DETAILS

### F.1 SA-MESH ARCHITECTURE

For the slot attention architecture, we closely follow Locatello et al. (2020b), in particular, we use the same CNN encoder and decoder as they use for CLEVR, except for the initial resolution of the spatial

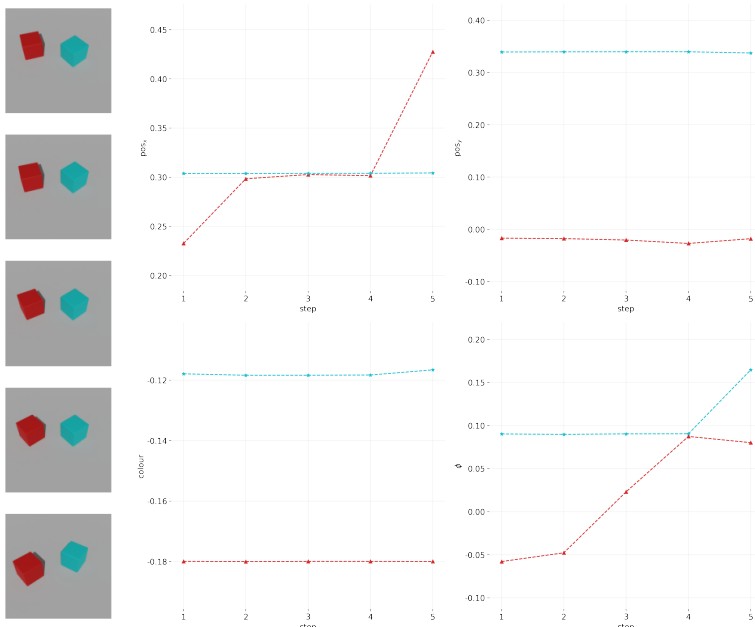

Figure 5: *(Left)* From top to bottom, the objects are perturbed as follows; Red: Downward perturbation so it sits at the same $x$ coordinate as the Blue cube (1-2), Rotates counterclockwise along the $z-$axis to be at the same orientation as the Blue cube (2-4), Further downward perturbation (2 times the displacement from step 1 to 2). Blue: Rotates counterclockwise along the $z-$axis (4-5). *(Right)* Note how the learned representations mapping correctly reflects the similar positions and rotations in the ground-truth, i.e., both by having properties of objects coincide at the same value, and by preserving the ratio of perturbations.

broadcast decoder with 3D shapes where we use $4 \times 4$ since we are dealing with $64 \times 64$ images. We use a slot size of 64 and always use $n + 1$ number of slots, where $n$ is the number of objects in the scene. We use 3 iterations for the recurrent updates in SA-MESH. For details concerning SA-MESH we follow Zhang et al. (2023). Additionally, we also truncate the backpropagation through slot updates as suggested by Chang et al. (2022) to improve training stability.

### F.2 DISENTANGLEMENT HEADS

SA-MESH outputs $n + 1$ slots that are of size 64, yet we need to project each of these slots to a $d-$dimensional space so we can leverage the disentanglement method from Ahuja et al. (2022a;b). We can simply achieve this projection by a single MLP, however, we decided to allocate more parameters for this projection and use $d$ separate projection heads mapping 64-dimensional vectors to $d$ separate scalars. This way identification of different properties will not affect one another due to model capacity constraints. We stack the layers shown in table 9 to obtain a projection head per each property. The same set of $d$ projections will be shared among all slots.

Table 9: Layers in a projection head for disentanglement.

| Layer | Input Size | Output Size | Bias | Activation |
|---|---|---|---|---|
| Linear (1) | 64 | 32 | True | ReLU |
| Linear (2) | 32 | 32 | True | ReLU |
| Linear (3) | 32 | 16 | False | ReLU |
| Linear (4) | 16 | 1 | False | ReLU |

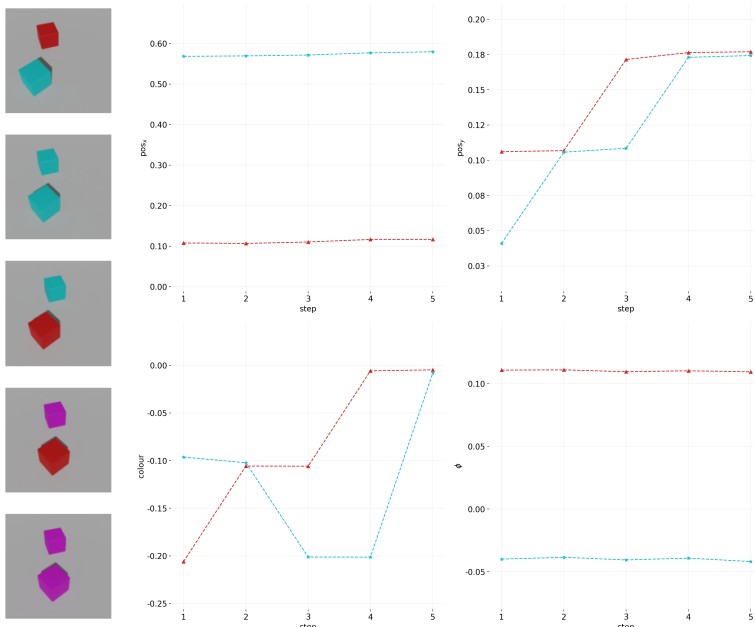

Figure 6: *(Left)* From top to bottom, the objects are perturbed as follows; Since the objects change color, let us call the red cube in the top frame to be object 1, and the blue one to be object 2. Object 1: The color is perturbed to be equal to object 2 (1-2), Moves to the right as its $pos_y$ is perturbed by 0.2 (2-3), The color hue is once again perturbed to become purple (3-4). Object 2: Moves toward right (perturbation in $pos_y$ by 0.2) to be at the same $y$ coordinate as object 1 (1-2), The color hue is *decreased* so now the colors of objects 1,2 are swapped (2-3). Moves by 0.2 in the $y$ direction to align with the other object once again (3-4), Change its color hue twice the previous color perturbation with the opposite sign to match the color of the other object (4-5). *(Right)* Red curves correspond to object 1, and the blue curves correspond to object 2. Again, notice the sections where the curves coincide, as well as the ratio of jumps in the properties, showing consistency of the learned representations with the ground-truth causal representation that gives rise to these observations.

## F.3 CONVNET BASELINE

As a baseline for injective scenarios, we use a ResNet18 (He et al., 2016) with an output width of 128 that is passed through `LeakyReLU` activation, which is then followed by $d$ linear projection heads (for the same reason we use separate disentanglement heads) that map the 128-dimensional output of the CNN encoder to $d$ separate 1-dimensional scalars that should correspond to the target $d-$dimensional space.

## F.4 TRAINING

For each $k$ and for each set of disentanglement target properties, we first train SA-MESH for 2000 epochs with a batch size of 64 for 2D shapes (as the images are $128 \times 128$), and a batch size of 128 for 3D shapes (since images are $64 \times 64$) on a single A100 GPU with 40GB of memory. We used a fixed schedule for the learning rate at $2 \times 10^{-4}$, and we used AdamW (Loshchilov & Hutter, 2017) with a weight decay of 0.01 along with $\epsilon = 10^{-8}, \beta_1 = 0.9, \beta_2 = 0.999$. SA-MESH was first solely trained by minimizing reconstruction error on the training set, then its disentanglement performance was reported on the test set for projection-based baselines (RP, PC, LR). Due to the high number of combinations of target disentanglement properties and $k$, we just trained SA-MESH for each configuration only once.

Unsupervised disentanglement with our method has an additional stage which takes the aforementioned pre-trained SA-MESH models and *jointly* minimizes the loss in equation 2. Note that at this stage, the SA-MESH model is *not* frozen, so the gradients flow through its network as well and help adjust the slot representations with the signal from the latent loss. Under *known perturbations*, we

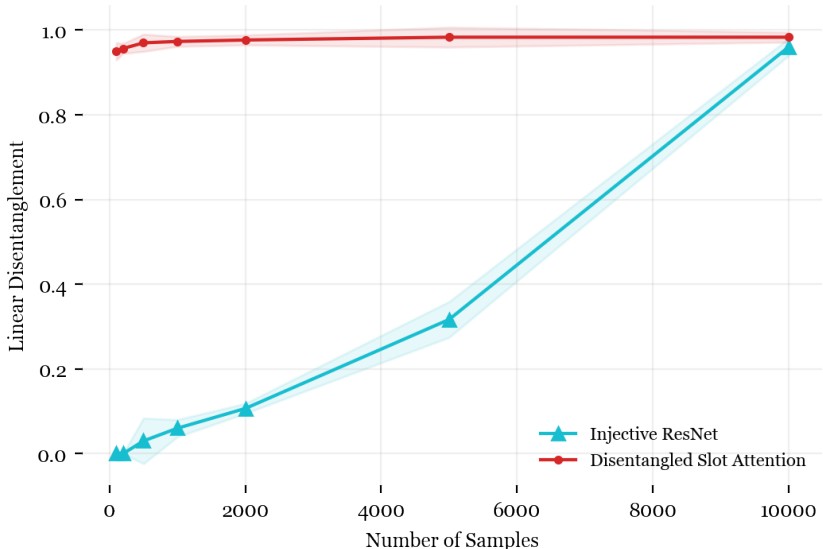

Figure 7: Comparing the disentanglement performance of an injective ResNet vs. our object-centric method based on the number of training samples. The dataset contains four 2D objects in which $\text{pos}_x, \text{pos}_y, \text{color}, \text{shape}$ can vary.

use the actual perturbations from the DGP as $\delta_t$ to guide the model via equation 2, however, under *unknown perturbations* setting, we replace all perturbations $\delta_t$ by a hyperparameter $C$ (see section 5). CNN baselines were trained similar to SA-MESH but for much shorter, i.e., 200 epochs, and usually converge very fast in less than 50 epochs.

## F.5 HYPERPARAMETER OPTIMIZATION

We started around the hyperparameters used by Locatello et al. (2020b) and Zhang et al. (2023) where applicable, and tuned on small subsets of the 2D shapes training data based on linear and permutation disentanglement metrics. We considered 5 values for the learning rate $[2 \times 10^{-3}, 2 \times 10^{-4}, 10^{-4}, 6 \times 10^{-5}, 2 \times 10^{-5}]$. Larger batch sizes were always better and we were only constrained by memory in the case of $128 \times 128$ images of the 2D shapes dataset. We considered 2 values $[0.1, 0.5]$ for $|C|$, the fixed value representing all unknown perturbations, and found $|C| = 0.1$ to perform better. We also considered slot sizes $[64, 128]$ on a small subset of the 2D shapes training dataset. Lastly we considered 9 combinations for the relative importance of latent loss and reconstruction loss when training the disentanglement heads, i.e., we considered all combinations of $w_{\text{latent}} \in \{1, 10, 100\}$, $w_{\text{recons}} \in \{1, 10, 100\}$, and found the combination of $w_{\text{recons}} = 100, w_{\text{latent}} = 10$ to strike the optimal balance between maintaining good reconstructions and allowing the slot representations to give rise to disentangled projections.

## F.6 DATASETS

**2D Shapes.** We use `pygame` engine (Shinners, 2011) for generating multi-object 2D scenes. Object properties in both datasets include $p_x, p_y$, color, shape, size, and rotation angle. Based on object properties, they are each rendered and placed on a white background (for the 2D dataset) or placed on a floor that is illuminated by source lights and is being visited from somewhere above the floor (for the 3D dataset), and then aggregated to produce $x_t$. We discretized the range of color hues in order to test the model's ability to obtain disentangled representations in the simultaneous presence of both continuous (position, size, and rotation angle) and discrete properties (color and shape).

In the 2D dataset:

- $p_x, p_y$ are generated randomly and uniformly in the $[0, 1]$ range, i.e., the boundaries of the scene, such that no two objects overlap and no object falls even partially outside the boundaries. Positional coordinates can be perturbed by any value in $[-0.2, 0.2]$.

- For color, we use HSV color representations and fix saturation (S) and value (V) at 0.6 and choose hue (H) from a set of values predefined before training (for instance $[0.0, 0.25, 0.5, 0.75]$). We adopted this 1-$d$ representation to be consistent and have each property be represented by a scalar. Additionally we wanted to test the model's capacity when dealing with mixed discrete properties. Also, training Slot Attention or SA-MESH with discrete colors is computationally advantageous since the model will not have to deal with reconstructing all colors. However it should be noted that HSV is a cylindrical geometry with color hues being the angular dimension which results in values that have a distance of 1.0 being exactly the same color (given a fixed saturation and value). That is why a list of color hues such as $[0.0, 0.33, 0.66, 1.00]$ would not work since 0.0 and 1.0 are the same color, yet our model interprets the difference as a perturbation with the amount of 1.0, which is clearly wrong. A change of color from color $i$ to $j$ where $i, j$ index the list of color hues $H$ would be provided to the model as a perturbation in the amount of $(H[j] - H[i])/|H|$, where $|H|$ denotes the number of colors in $H$.

- Shape is also clearly discrete and is selected at random uniformly from the following set of shapes $S = \{\text{circle}, \text{square}, \text{triangle}, \text{heart}, \text{diamond}\}$. Note however, that the effects of perturbations need to be visible in the pixel space, and we should be wary of the disentanglement target properties; for instance if the rotation angle $\phi_t$ is a property we aim to disentangle with perturbations, then we should exclude *circle* from the set of possible shapes since a circle does not reflect in the pixel space the angle perturbations. A shape transformation from shape $i$ to $j$ where $i, j$ index $\mathcal{S}$, would be provided to the model as a perturbation with the amount of $(j - i)/|\mathcal{S}|$.

- Size is a continuous property in the range $[0.12, 0.24]$ of the height or width of the image which is 1. It can be perturbed by any amount in $[-0.02, 0.02]$.

- Rotation angle is also another continuous property in $[0, \pi/4]$. Similar to color hues, since this property is also angular, we have limited the range not to encounter situations that appear the same in the pixel space but have very different rotation angles (a square that is rotated $\pi/2$ clockwise seems unaltered, or $\pi/4$ and $3\pi/4$ rotations both look the same for a square.). Angular perturbations can vary in the $[-0.2, 0.2]$ range.

We generate samples in pairs corresponding to $t, t + 1$. For fully dense perturbations, we generate $k$ vectors of dimension $d$, where $k$ is the number of objects. We repeat the generation until the conditions of non-overlapping objects and non-identifiability are met, i.e., no two objects at either $t$ or $t + 1$ should overlap (before and after the perturbations), no object should fall in whole or partially out of the scene, and no two objects should be perturbed by $d$-dimensional offsets that are closer than some $\epsilon$. The last condition is necessary for fully dense perturbations as otherwise the matching has no way of distinguishing which perturbation to assign to which object since the matching solely relies on the *difference* between $t, t + 1$. For fully sparse perturbations, we are not constrained by the latter, and we only need to choose perturbations that do not push the chosen object out of boundaries, or make it overlap with another object. For any experiment we can have a subset of $\{\text{pos}_x, \text{pos}_y, \text{color}, \text{shape}, \text{size}, \text{rotation angle}\}$ as the properties we wish to disentangle by observing perturbations in the pixel space, and we call them *disentanglement target properties*. In the generation process, any non-target property will be fixed for all objects in the whole dataset to avoid introducing unwanted variance to the disentanglement of target properties, i.e., if we choose $\{\text{pos}_x, \text{pos}_y, \text{color}, \text{shape}, \text{rotation angle}\}$ as target properties, then all the objects in all samples would have the same fixed size. Lastly, we can choose to make a DGP injective or not. If we choose to make a DGP injective, we would index the objects and choose a property to be set for objects according to the indices, i.e., we can choose to make the DGP injective by color; Suppose $k = 4$ and the list of our color hues is $[0.0, 0.25, 0.5, 0.75]$. We would color the objects, which are now ordered according to some index set $\mathcal{I}$, according to $\mathcal{I}$. Non-target properties (excluding the injectivity imposing property) will again be kept fixed for the whole dataset, and target properties are generated according to fully dense or fully sparse perturbation schemes. The perturbations to all properties are *signed*, and this is especially crucial for discrete properties such as shape. The reason is that disentanglement is achieved through observing *relative* distances in the pixel space, and having only positive or only negative perturbations deprives the model of having a reference for each property.

For training we generate 1000 pair per target property such that the model on average sees at least 500 samples for either positive or negative perturbations to each property, i.e., if we choose $\{\text{pos}_x, \text{pos}_y, \text{color}\}$ as target properties, we will generate 3000 samples for training. The validation and test sets always have 1000 samples. For the 2D dataset, we generate $128 \times 128$ images for better visual quality that is not distorted due to artifacts caused by perturbations. We then normalize and clip the image features (RGB values) to be in $[-1, 1]$ range.

**3D Shapes.** For generating the 3D datasets we leverage `kubric` library (Greff et al., 2022) to obtain realistic scenes which we can highly customize. Objects sit on a floor, a perspective camera is situated at $(2.5, 0, 3.0)$ and looks at $(0.0, 0.0, 0.0)$. Directional light illuminates the scene from $(1.0, 0.0, 1.0)$ towards the center. The set of possible target properties are similar to 2D shapes, and the range of properties in which each object is spawned is as follows:

- $\text{pos}_x, \text{pos}_y$ are generated randomly and uniformly in the $[-1.5, 1.5]$ and $[-1.0, 1.0]$ ranges respectively, such that no two objects overlap and no object falls even partially outside the boundaries. Note however, by overlap we mean that objects are spawned such that they do not mutually fill a volume in the 3D space, and we only prevent such occurrences, but we *do* allow *occlusions* from the perspective of the camera, which adds to the complexity of this synthetic dataset. $\text{pos}_z$ is never a disentanglement target property and is always set such that objects sit on the floor (except when rotated). The reason for fixing the $z$ coordinate is that any possible perturbation to the 3D coordinates is always going to be interpreted on a 2D scene that is observed by a camera that is placed somewhere above the floor. Therefore, introducing a third coordinate in the DGP and target properties has no point. Positional coordinates can be perturbed by offsets in $[-0.3, 0.3]$ range.
- color is similarly parameterized by a scalar in HSV format as in 2D.
- Shape can be any of $\{\text{sphere}, \text{cube}, \text{cylinder}, \text{cone}\}$. Again, since the effects of perturbations need to be visible in the pixel space, we will not use spheres if the rotation angle $\phi$ is a property we aim to disentangle with perturbations, as sphere rotations do not reflect in the pixel space. For rotation in the 3D space we choose the $z$-axis as the axis of rotation so that angular perturbations are maximally visible (w.r.t. the perspective camera's location).
- Size is a continuous property in the range $[0.3, 0.7]$ and can be perturbed by offsets in $[-0.15, 0.15]$ range.
- Rotation angle follows the convention of 2D shapes DGP, except that the rotations are around the $z$-axis for better visual quality.

Since `kubric` already generates high fidelity images, we use $64 \times 64$ images to lower the computational burden of SA-MESH autoencoder. The number of samples is always fixed at $20,000$ regardless of the target properties since the 3D dataset is more complex. We use a similar transformation as in 2D, and normalize and clip the image features (RGB values) to be in $[-1, 1]$ range.

### F.7 LIMITATIONS

Our study focuses on showing when disentanglement is possible when treating object-centric environments as a set of representations instead of fixed-size vectors. We have analyzed the performance of our model comprehensively on two synthetic datasets that are relatively limited in capturing the complexities of real-world scenarios. Yet, we believe and showed such analysis is a necessary first step to identify the intricacies involved in making our algorithm work. Our analysis has been limited in a number of directions. First, while we do consider a wide range of continuous and discrete properties to be disentangled, the number of objects we use is rather low, which ideally should be scaled to real-world scenes containing more objects. Second, although our experiments include artifacts related to occlusion, depth, and lighting, in all of our experiments we simplify the problem by having the objects situated on homogeneous backgrounds, whereas real-world scenes would comprise more complex backgrounds. Such decisions were mainly due to (1) generating datasets of size more than 5k for each combination of properties being a computationally heavy task on its own, (2) training SA-MESH from scratch for each combination of properties and number of objects would quickly add up as each training takes $\sim 12$ hours on a single A100 GPU to achieve nice reconstructions, (3) details related to the background and the number of objects are tangential to the focus of this study, which is to demonstrate how to disentangle the causal factors in an object-centric environment.

