# OpenReview forum: "Object centric architectures enable efficient causal representation learning"
_ICLR.cc/2024/Conference — ICLR 2024 poster_

### Official Review · Reviewer_HmRq · 2023-11-01

**Soundness:** 4 excellent
**Presentation:** 3 good
**Contribution:** 3 good
**Rating:** 8
**Confidence:** 5

**Summary:**

This work addresses the issue of non-identifiability in causal representation learning models using object-centric approaches. The authors show that using sparse transitions between pairs of images and models that exploit such regularities can allow to solve the responsibility problem when attempting to invert a data-generation process in the image domain. Specifically, the exploit the fact that object centric models partition images into objects to reduce the number of interventions needed in order to identify the properties that correspond to each object thus allowing them to be identified. To encourage identifiability the authors introduce a penalty that seeks to keep all slots in the object-centric representation constant for both images, which can them only vary for the object that was perturbed. The authors then show that this leads to an increase in the ability of downstream models to predict the different properties of the objects in a scene.

**Strengths:**

1. The paper is well motivated based on the limitations of previous work in causal representations learning and object-centric approaches.
2. The authors correctly justify their approach with easy to follow but formal statements (or propositions) that theoretically ground their approach.
3. While maybe a bit light on experiments, I believe that for their particular purposes the results substantiate their results, with one caveat that I will address later.

**Weaknesses:**

1. The authors do however miss the opportunity to connect their research with previous work on disentanglement learning in a more systematic way. Indeed their approach of encouraging sparse transitions has already been explored in disentangled models (albeit not object-centric ones) in Klindt et al., (2021) and Montero et al., (2022). and thus these are both relevant work that should be discussed (though I don't think a comparison with these methods is required).
2. While I don't have an issue with the evaluation per se, I do think that the scores of the models should be separated by properties as well as averaged over all of them (as is currently presented according to my understanding).



Refs:
[1] Klindt, D., Schott, L., Sharma, Y., Ustyuzhaninov, I., Brendel, W., Bethge, M., & Paiton, D. (2020). Towards nonlinear disentanglement in natural data with temporal sparse coding. arXiv preprint arXiv:2007.10930.
[2] Montero, M. L., Bowers, J. S., Costa, R. P., Ludwig, C. J., & Malhotra, G. (2022). Lost in latent space: Disentangled models and the challenge of combinatorial generalisation. arXiv preprint arXiv:2204.02283.

**Questions:**

My only question is related to how good the prediction of each individual object property is. My understanding is that the prediction scores are averaged over all properties, but previous research has show that some properties are easier to predict than others (eg. rotation is harder than position). Thus, I would like to see the scores for these different properties in isolation.

---

> ### Author Response · Authors · 2023-11-19
>
> We would like to thank reviewer 9Arp for your valuable review, and we appreciate the time and effort you invested in evaluating our work.
>
> ## Clarifications concerning the weaknesses
>
> **Weakness 1:**
> We thank reviewer HmRq for bringing these works to our attention, however, we would like to emphasize that we are not claiming contribution to the sparse transitions in disentanglement, we are simply building on top of Ahuja et al. [1] as a modeling choice. Other choices are possible like the ones mentioned here by reviewer HmRq. We acknowledge that we should make this modeling choice more clear and mention the possibility of using the aforementioned works as well.
>
> **Weakness 2:**
> We thank reviewer HmRq for bringing up this important point. While we acknowledge the usefulness of the additional insight provided by reporting per-property disentanglement scores, we would like to clarify that the presented scores aren't exactly average of per-property disentanglement scores. Intuitively, if say, we aim to disentangle $(p_x,p_y,\text{colour},\phi)$, and the model has only been able to separate just one property from the rest, then the score would be about $0.25$ or $\frac{1}{d}$, where $d$ is the number of target properties. If two properties are separated from the rest, then the score would be about $0.5$, and so on. So a score $0.95$ indicates that almost all properties are identified to a high degree. We hope this clarifies how the disentanglement scores work, and additionally, we will provide per-property disentanglement scores as well before the end of discussion period. Please expect another reply on this matter.
>
>
> ----
> ### **Answers to Questions:**
>
> 1. Again, we thank the reviewer for bringing up this important observation. Above, we explained in more detail how these scores can be interpreted. However, we acknowledge that we have observed similar patterns to those that reviewer HmRq mentions. In particular, position properties are always far easier separated than the rest of the properties. We have also particularly witnessed more challenges with discrete properties, i.e., colour and shape, while rotation and size have also been fairly easy to disentangle. In summary, approximately we find the following ordering in increasing difficulty of separating the property:
>
> $p_x,p_y < \text{size}, \text{rotation} < \text{colour}, \text{shape}$
>
> However, we do believe there can be additional insight by reporting per property disentanglement scores, and we will provide those before the end of the discussion period.
>
> We thank the reviewer again for raising these interesting points.
>
>
>
> ### References
> [1] Weakly supervised representation learning with sparse perturbations (Ahuja et al.)

---

> > ### Comment · Reviewer_HmRq · 2023-11-20
> >
> > I thank the authors for their clarifications.
> >
> > It is now clear to me how the scores are computed. I also look forward to the extra results involving disentanglement scores of each individual property which I think hold significant value as they can point to future directions of research. Otherwise I am satisfied with their work and will be keeping my score.

---

> > > ### Author Response · Authors · 2023-11-22
> > >
> > > We thank the reviewer for their kind comment. We conducted additional experiments on 3D shapes in the presence of 2,3,4 objects with the following target properties $(p_x,p_y,\text{colour},\text{rotation})$ to assess the individual property disentanglement scores, and the findings show that the originally presented scores indeed reflect the individual property disentanglement as well, and basically we don't see any discrepancy with the originally presented numbers. I.e., the scores for this case study are very close ($\pm0.01$) to those of table 7 in the appendix. However, different properties do not get disentangled at the same time, with discrete ones (colour in this case) still being the more challenging properties, however, once disentangled, all properties are very well and equally disentangled.
> > >
> > > However, if the reviewer has further suggestions, we could set up a more systematic evalution of individual property disentanglement and assess the behavior on more settings (2D for instance) and other combinations of target properties.
> > >
> > > Thanks again for your insightful comments.

---

### Official Review · Reviewer_T5fz · 2023-11-03

**Soundness:** 3 good
**Presentation:** 3 good
**Contribution:** 2 fair
**Rating:** 6
**Confidence:** 3

**Summary:**

The paper studies causal representation learning in the context of multi-object scenes, challenging the assumptions commonly held in previous work on single-object images (such as injectivity). The paper proposes a new approach for disentangling object properties via weak supervision under sparse perturbations and assesses its effectiveness on synthetic datasets.

**Strengths:**

-	The study effectively bridges recent advances in causal representation learning with object-centric learning, both from theoretical and methodological perspectives. The theoretical analysis identifies and discusses the issue of non-identifiability (e.g., non-injectivity) in multi-object scenarios. The method part nicely adapts the weakly-supervised approach from disentangled/causal representations to multi-object contexts.
-	The paper is very well written and enjoyable to read.

**Weaknesses:**

- Insight: While technical sound, the proposed method itself appears not as interesting. If I understand correctly, the method is essentially nothing but adding a slot matching step to the existing weak-supervised disentangled representation learning.
- Positioning: The claim of being the first to `achieve disentangled representations in environments with multiple interchangeable objects` is overstated. Similar objectives have been pursued in recent studies, e.g., Block-Slot Representation [1]. Slot matching has also been considered in recent work [2]. A more thorough positioning in the literature about multi-object scenes might be needed.
- Theory: the proof sketch for Proposition 2 appears less than compelling in its current form. Part of my doubt is attributed to the use of identical objects. It would strengthen the paper if the author could prove Proposition 2 in more common scenarios.

[1] Neural Systematic Binder, ICLR'23 \
[2] Causal Triplet, CLeaR'23

------------------------------------------------

Post-rebuttal Review:

The response from the authors has addressed the primary concerns I had previously. I thus raise the rating from 5 to 6. \
The theoretical part is definitely a solid contribution, whereas the insight and positioning of the current manuscript remain somewhat weak. \
I would recommend the authors to consider incorporating their rebuttal into the final version of their paper.

**Questions:**

- It's stated in Sec4 that the object-centric model requires k times fewer perturbations. Is there any empirical evidence to support this claim?
- For slot matching, is it necessary to solve an expensive optimization problem (Eq13)? How does it perform compared to a simple initialization trick (e.g, using the slot of the 1st image to initialize that of the 2nd)?
- Could the authors provide a more detailed explanation on `simply add a projection head, that is trained by a latent space loss`? Why is the projection head needed on top of the object-centric encoder?

---

> ### Author Response · Authors · 2023-11-19
>
> We would like to thank reviewer T5fz for your valuable review, and we appreciate the time and effort you invested in evaluating our work, and we are glad that you have enjoyed reading our work.
>
> ----
> ## Clarifications concerning the weaknesses
>
> **Weakness 2**:
> We appreciate the reviewer's observation regarding our positioning and would like to address this concern. We acknowledge the need for a more explicit emphasis on the positioning of our work. While [1] explores a similar problem **empirically**, we want to highlight that our contribution lies in providing a **provable** disentanglement of object-centric representations, followed by strong experimental results against strong baselines. As outlined in our paper (please refer to the footnote on page 3), [1] does not address the identifiability of their approach, and our work fills this crucial gap by offering formally accompanied by empirical results. We thank the reviewer for drawing attention to this aspect and will make necessary revisions to emphasize this point more effectively.
>
> Regarding slot matching in [2], we recognize the presence of some form of slot matching in their work, but it is essential to clarify that our primary contribution is not the matching algorithm itself. Instead, our focus is on identifying and addressing challenges that arise from objects violating standard assumptions for latent variable identification. We demonstrate how leveraging object-centric architectures can effectively handle these challenges, showcasing the sample efficiency of our method, as highlighted in the appendix. In summary, even though our matching approach differs from [2], it is not also a central contribution of our work. Moreover, [2] provides a benchmark for causal representation learning, and the matching used in [2] is for intervention modeling.
>
> **Weakness 1**:
> In response to the concern raised, we would like to emphasize that our work addresses the previously unexplored territory of "provable" disentanglement of object-centric representations. Our approach connects identifiability results from prior work to real-world scenarios with multisets, filling a crucial gap in the literature. Furthermore, we believe the simplicity of our model adds to its strength. By demonstrating the effectiveness and consistency of a straightforward model across various settings, we aim to establish it as a reliable and strong approach for generalizing previous results to realistic environments.
>
> **Weakness 3**:
>
> We appreciate the reviewer's feedback on the proof sketch for Proposition 2. We would like to highlight that our goal has been to provide intuition without excessive mathematical formalism in this section, and for a more general proof that does not require identical objects, we refer the reader to see Hayes.
>
> ----
> ### **Answers to Questions:**
>
> 1. Yes, empirical evidence supporting the claim in Sec. 4 about the object-centric model requiring $k$ times fewer perturbations is provided in Figure 7 in Appendix E.3. The observed benefits are not only quite significant but also surpass the $k$ times reduction, reinforcing the practical advantages of our proposed method in the finite sample regime.
>
> 2. The optimization problem in Eq. 13 for slot matching is computationally inexpensive, requiring only finding the minimum of a small array (same size as the number of objects).
> Moreover, as mentioned prior to Eq. 13, we do use the mentioned initialization trick and use slots obtained at $t$ to initialize those of $t+1$. However, this trick alone is not enough. We have observed that in simplistic 2D experiments with few objects and few target properties this trick works fine, however, in more complex settings, we observe that the order of extracted objects at $t,t+1$ is not quite preserved, and such occasional mismatch causes wrong training signals significantly preventing the optimization from finding disentangled solutions. Therefore, the use of this trick alone in more complex settings results in significantly suboptimal disentanglement, leaving some harder properties entangled altogether.

---

> > ### Author Response · Authors · 2023-11-19
> >
> > 3. The need for a projection head on top of the object-centric encoder is explained in two possible contexts. If the question pertains to using low-dimensional slots for direct reconstruction, we clarify that such representations are insufficient for reconstruction, leading to autoencoding failures. Alternatively, if the question is about why we need to project high-dimensional slot representations, we highlight the necessity of enforcing the latent space penalty on low-dimensional latents (see eq. 1). The reason is that we enforce the penalty through proxies on structural knowledge (low-dimensional offsets), and this penalty cannot be directly applied to slots. A similar reasoning motivated the choice of our baselines (Sec. 7). In summary, the autoencoder loss operates on 64-dimensional slot representations, while the latent loss works with {2,3,4,5}-dimensional projections of those 64-dimensional representations, requiring the additional projection step.

---

### Official Review · Reviewer_9Arp · 2023-11-08

**Soundness:** 3 good
**Presentation:** 3 good
**Contribution:** 2 fair
**Rating:** 6
**Confidence:** 3

**Summary:**

The authors focus their attention on disentanglement in an object centric environment. Authors formally introduce the challenges that may arise when multiple objects are present in the observations, especially when objects can be identical. Authors also show the effect experimentally by plotting the achieved MCC score. Authors combine the ideas of object wise partition and Ahuja et al.'s weak supervision disentanglement to solve the aforementioned problem with a claim that fewer perturbations are needed in such procedure due to shared latent spaces in the objects. Experiments are conducted with a single set of latent variables shared across multiple objects in the scene, with 2D and 3D synthetic dataset. Authors deal with both discrete and continuous latents. Experiments are conducted against a strong baseline.

**Strengths:**

- Clear communication of proposition and idea
- Well thought through related work sections and explanations
- Always felt I understood what authors are trying to say
- Simple yet elegant idea to solve the multiple object disentanglement problem
- Strong baseline for experimentation

**Weaknesses:**

- I feel the experimental setup was quite limiting for the proposed claim
- How do we handle multiple set of latents? What happens when there are shared and not-shared latents across this set how does the disentanglement look like?
- Authors mentions limitations on number of objects and background although I would like to see the effect on some real life dataset (maybe can be constructed with multiple colored mnist images in the scene?)

**Questions:**

Minor Edits -
 - Double "them" in Proposition 2 Proof Sketch

---

> ### Author Response · Authors · 2023-11-19
>
> We would like to thank reviewer 9Arp for your valuable review, and we appreciate the time and effort you invested in evaluating our work.
>
> Regarding the following,
> > Experiments are conducted with a single set of latent variables
> >
> we would like to clarify and highlight that our experiments are *not* carried out with *only* a single set of latent variables. While we deal with shared set of latents across objects, we show the effectiveness of our approach with varying set of latents, i.e., disentangling only $(p_x,p_y)$, or $(p_x,p_y,\text{colour},\text{shape})$, or $(p_x,p_y,\text{colour},\phi)$ (Please see the appendix for all the combinations we have tried.)
>
> ## Clarifications concerning the weaknesses
>
> **Weakness 1**: We would appreciate it if reviewer 9Arp could elaborate on this, and kindly let us understand what they have in mind, and what kinds of experiments would be interesting to be added to our work?
> We would like to highlight two aspects of this work in response to this concern. First, the experiments conducted in this work are significant steps toward realistic applications of causal representation learning (CRL), and they are significantly more challenging than the experiments of the current literature in causal representation learning that has mostly focused on very toy settings with one or two circles [1,2,3,4]. Secondly, an important aspect of evaluating such approaches in CRL is access to ground-truth which is usually achievable through synthetic datasets, yet we have tried to bridge the gap as much as possible by using synthetic samples that are realistic (CLEVR-style). The synthetic nature of our dataset provides extra handle on stress testing our method, and allows us to quickly identify the failure modes, necessary practical assumptions, etc. However, we strive to improve our work and are keen to learn more about the experiments that reviewer 9Arp believes would strengthen this work.
>
> **Weakness 2**: In the presence of non-shared latents, the method still applies in its current form, however, unsurprisingly, we lose the benefit of requiring $k$ fewer perturbations. Good examples illustrating this point can be found in video games. There are properties such as characters moving around that are shared across all of them, and such properties are far easier to be identified due such knowledge being reused over and over. However special characters in the game might have superpowers unique to them, and consequently, learning about such special abilities is only possible through observing that one special character, without any knowledge transfer from the features of other characters who do not share this property.
>
> **Weakness 3**: We thank reviewer 9Arp for the suggestion. We would like to highlight that despite the differences between the proposed dataset and our experiments at the surface level, these experiments are actually not qualitatively different **in terms of disentanglement**. The only difference is **perception** (i.e., slot attention + the preceding vision model), not the disentanglement approach. We know from other works using slot attention ([5,6,7,8]), that the perception works just fine in far more realistic settings than our shapes experiments and the proposed MNIST dataset. The proposed disentanglement method is contingent on good slot representations, and given the success of slot attention, we think there is limited qualitative difference **in terms of disentanglement** between our experiments and the reviewer's proposed dataset.
> [We are reading the question as meaning MNIST == realistic dataset, and asking us to create a dataset with multiple coloured MNIST digits with varying sizes, rotations etc.]
>
>
> ### References
> [1] Weakly Supervised Representation Learning with Sparse Perturbations (Ahuja et al.)
>
> [2] Interventional Causal Representation Learning (Ahuja et al.)
>
> [3] Partial Disentanglement via Mechanism Sparsity (Lachapelle et al.)
>
> [4] Additive Decoders for Latent Variables Identification and Cartesian-Product Extrapolation (Lachapelle et al.)
>
> [5] Bridging The Gap To Real-world Object-Centric Learning
>
> [6] SlotFormer: Unsupervised Visual Dynamics Simulation With Object-centric Model
>
> [7] Conditional Object-Centric Learning from Video
>
> [8] SAVi++: Towards End-to-End Object-Centric Learning from Real-World Videos

---

### Author Response · Authors · 2023-11-23

As the discussion period deadline is closing in, once again, we would like to express our sincere gratitude for the time and effort that all reviewers have invested in reviewing our work, we understand the constraints of time in the peer review process and greatly appreciate your contributions thus far.

Given that the discussion deadline is fast approaching and we won't be able to provide further replies after that point, we want to highlight that we tried to address your concerns carefully, and we kindly ask reviewers **9Arp** and **T5fz** to revisit our paper based on the provided clarifications at your earliest convenience.


If our responses have effectively clarified any questions or uncertainties, we would greatly appreciate your consideration in improving your score. Your insights are pivotal, and we want to ensure that your evaluations are based on the most comprehensive information available.

We want to emphasize our commitment to continuous improvement of our work. If there are any lingering concerns or if any of the reviewers have additional suggestions, please let us know. Your feedback is crucial to refining our manuscript, and we are eager to incorporate any further insights you may have.

Sincerely,

The Authors

---

### Meta-Review · Area_Chair_Fwqh · 2023-12-06

**Metareview:**

The paper addresses the setting of causal representation learning in the presence of images containing multiple objects. Using insights from the object-centric model literature, specifically Slot Attention, the authors address limitations of previous causal models, proposing an approach based on sparse perturbations. The reviewers generally found the paper clear and seem to agree that the paper provides a useful contribution in linking causal representation learning with object-centric models. The reviewers also raised various somewhat minor concerns pertaining to novelty and the empirical evaluation, which the authors largely addressed through their responses.

**Justification For Why Not Higher Score:**

While the reviewers were in agreement on acceptance, the proposed approach is not massively novel, nor is the empirical evaluation more broadly inclusive of more complex datasets / settings. That is, this paper is relevant to the specific communities of causal representation learning and object-centric models, but it may not warrant a spotlight / oral.

**Justification For Why Not Lower Score:**

The paper connects two areas of the literature (causal representation learning and object centric models). The authors are in agreement that the significance of this paper’s contribution warrants publication.

---

### Decision · Program_Chairs · 2024-01-16

Accept (poster)